# Glocal Smoothness:
# Line search and adaptive sizes can help in theory too!

**Curtis Fox**                                     *curtfox@cs.ubc.ca*
*Department of Computer Science*
*University of British Columbia*

**Aaron Mishkin**                                  *aaronpmishkin@gmail.com*
*Department of Computer Science*
*Stanford University*

**Sharan Vaswani**                                 *vaswani.sharan@gmail.com*
*School of Computing Science*
*Simon Fraser University*

**Mark Schmidt**                                   *schmidtm@cs.ubc.ca*
*Department of Computer Science*
*University of British Columbia*
*Canada CIFAR AI Chair (Amii)*

**Reviewed on OpenReview:** *https://openreview.net/forum?id=be9PdukwEL*

## Abstract

Iteration complexities for optimizing smooth functions with first-order algorithms are typically stated in terms of a global Lipschitz constant of the gradient, and near-optimal results are then achieved using fixed step sizes. But many objective functions that arise in practice have regions with small Lipschitz constants where larger step sizes can be used. Many local Lipschitz assumptions have been proposed, which have led to results showing that adaptive step sizes and/or line searches yield improved convergence rates over fixed step sizes. However, these faster rates tend to depend on the iterates of the algorithm, which makes it difficult to compare the iteration complexities of different methods. We consider a simple characterization of global and local ("glocal") smoothness that only depends on properties of the function. This allows upper bounds on iteration complexities in terms of iterate-independent constants and enables us to compare iteration complexities between algorithms. Under this assumption it is straightforward to show the advantages of line searches over fixed step sizes and that, in some settings, gradient descent with line search has a better iteration complexity than accelerated methods with fixed step sizes. We further show that glocal smoothness can lead to improved complexities for the Polyak and AdGD step sizes, as well other algorithms including coordinate optimization, stochastic gradient methods, accelerated gradient methods, and non-linear conjugate gradient methods.

## 1 Setting the Gradient Descent Step Size

Machine learning models are typically trained using numerical optimization algorithms. One of the simplest algorithms used is gradient descent (Cauchy et al., 1847), which on iteration $t$ takes a step of the form

$$w_{t+1} = w_t - \eta_t \nabla f(w_t), \tag{1}$$

for some positive step size $\eta_t$. The simplest way to set the step size $\eta_t$ is to use a constant value throughout training. Other methods have also been proposed to obtain faster convergence in practice, including line

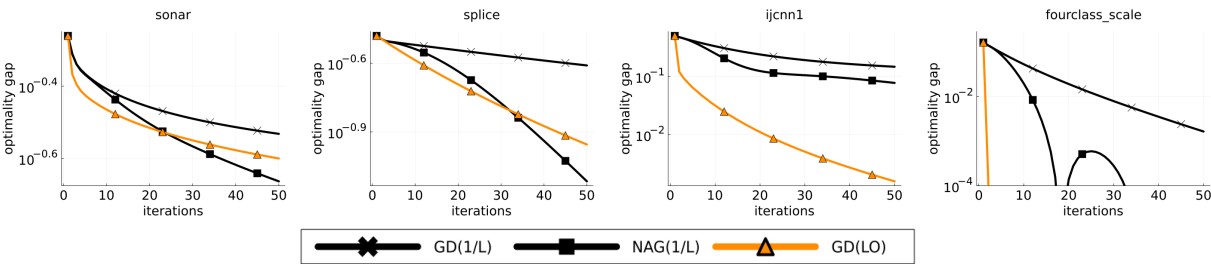

Figure 1: Logistic regression on 4 datasets showing the optimality gap throughout training for gradient descent (GD) and Nesterov's accelerated gradient (NAG) method with a fixed step size of $\frac{1}{L}$, along with gradient descent with line optimization (LO). From left to right are a problem where GD(LO) has a similar rate to GD(1/L), a problem where GD(LO) is converging faster than GD(1/L), and two problems where GD(LO) is converging faster than NAG(1/L). These experiments were generated using the code and experimental setup of Shea & Schmidt (2024).

searches (Armijo, 1966) or adaptive step sizes such as the Polyak step size (Polyak, 1987). To analyze the convergence rate of gradient descent we typically assume global $L$-smoothness, meaning that the objective $f : \mathbb{R}^d \to \mathbb{R}$ is assumed to have a globally Lipschitz continuous gradient.

**Definition 1** (Global $L$-Smoothness). A function $f$ has an $L$-Lipschitz continuous gradient if $\forall u, v \in \mathbb{R}^d$,

$$\|\nabla f(u) - \nabla f(v)\| \le L \|u - v\|.$$

Unfortunately, the step sizes that give optimal theoretical convergence rates and those that work well in practical applications may not always be the same. To illustrate the differences between theory and practice, contrast the fixed step size $\eta_t = \frac{1}{L}$ with using line optimization (LO) to minimize the function,

$$\eta_t \in \arg\min_{\eta} f(w_t - \eta \nabla f(w_t)). \tag{2}$$

We refer to gradient descent with $\eta_t = \frac{1}{L}$ as GD(1/L) and gradient descent with $\eta_t$ set using (2) as GD(LO). Standard analyses of GD(1/L) and GD(LO) give identical worst-case theoretical convergence rates for $L$-smooth functions (Karimi et al., 2016). But in practice, GD(LO) often performs much better than its worst case behaviour. Indeed, GD(LO) typically converges much faster than GD(1/L). As a further illustration of the gap between worst-case convergence rates and practical performance, consider GD(LO) compared to Nesterov's accelerated gradient (NAG) method with a step size of $\frac{1}{L}$ (Nesterov et al., 2018) which we call NAG(1/L). Although GD(LO) has a slower worst-case theoretical convergence rate in terms of $L$, GD(LO) often converges faster in practice (see Figure 1).

One reason GD(LO) can converge faster than GD(1/L) and NAG(1/L) in practice is that it can use much larger step sizes than $\frac{1}{L}$ which may lead to more progress. These larger step sizes are possible because, in a region around each $w_t$, the Lipschitz smoothness assumption (1) may hold with a smaller constant $L$. Many works have analyzed first-order methods under local measures of $L$ (see Section 2). However, these analyses generally depend on the iterates $w_t$ of the algorithm, making it difficult to compare convergence rates between algorithms since different algorithms will take differing paths during training and thus have different local $L$ values. Thus, it is hard to use these assumptions to theoretically justify why GD(LO) can significantly outperform NAG(1/L).

In Section 3, we introduce a *glocal* ("global-local") smoothness assumption that augments global smoothness with a measure of local smoothness near the solution. A function is $(L, L_*, \delta)$-glocal smooth if it is globally $L$-smooth and locally $L_*$-smooth when the sub-optimality is at most $\delta$. This characterization of global and local smoothness allows theoretical analyses to reflect, for example, that GD(LO) may have a faster worst-case convergence rate than GD(1/L) locally near the solution. In Section 3 we consider iteration complexity bounds for GD(LO) under strong convexity and the glocal smoothness assumption. The iteration complexity is the number of iterations for an algorithm to reach an accuracy of $\epsilon$, and under glocal smoothness we

show that GD(LO) has a better worst-case iteration complexity than GD(1/L) whenever $L_* < L$. While it is possible to show an improved iteration complexity of GD(LO) over GD(1/L) under previous local smoothness assumptions, the advantage of analyses based on glocal smoothness is that they only depend on the properties of the function being optimized, and not on the precise iterates taken by the algorithm. Thus, under glocal smoothness we can also compare the worst-case iteration complexities of GD(LO) and NAG(1/L). Indeed, we give conditions under which GD(LO) has a significantly better iteration complexity bound than NAG(1/L) in terms of $L$ and $L_*$.

Our analysis in Section 3 focuses on LO since it leads to simple and elegant results. However, in most cases LO is not practical. In Section 4 we discuss how similar results hold under practical step size selection strategies like Armijo backtracking. Alternate step sizes that can work well in practice such as the Polyak step size (Polyak, 1987) and the AdGD step size (Malitsky & Mishchenko, 2020) are discussed in Section 5 under an iterate-based variant of the glocal assumption. Section 5 also discusses relaxations of our strong convexity assumption such as assuming the Polyak-Łojasiewcz (PL) inequality or only assuming convexity. In Section 6, we show that glocal smoothness yields improved complexity bounds for other algorithms including coordinate descent, stochastic gradient descent, NAG, and non-linear conjugate gradient.

We close this introduction by highlighting a subtle but important point. All the iteration complexity bounds we show under glocal smoothness hold simultaneously *for all values of $\delta$* (and their corresponding $L_*$ values). Thus, we obtain the tightest bound by minimizing over values of $\delta$. We show how to do this explicitly for logistic regression in Section 3.7). We believe that it is important in practice to be able to adapt to any value of $\delta$, and hope that this work leads theoreticians to consider algorithms with this adaptability.

## 2 How much do Line Search and Local Smoothness help?

It may seem possible that the performance of GD(LO) may be explained by a better analysis under global smoothness. Indeed, recent work (de Klerk et al., 2017) gives tight rates for GD(LO) that are faster than for GD(1/L). However, these rates do not explain why GD(LO) can converge much faster than NAG(1/L).

Many works define a notion of local smoothness based on the iterates $w_t$ (Zhang & Yin, 2013; Scheinberg et al., 2014; Grimmer, 2019; Mei et al., 2021; Berahas et al., 2023; Lu & Mei, 2023; Orabona, 2023; Mishkin et al., 2024). These local smoothness conditions depend on the lines between successive iterates $w_{t-1}$ and $w_t$, balls around the $w_t$, the convex hull of the $w_t$, or $f(w_t)$ sub-level sets. Consider using $L(w_t)$ to denote one of these measures of local smoothness at iteration $t$. In many cases, we can modify existing algorithms to adapt to the problem by allowing larger step sizes that exploit the local smoothness $L(w_t)$. This allows adaptive algorithms to obtain convergence rates depending on the sequence of $L(w_t)$ values that are faster than iterate-agnostic rates that depend on the global maximum $L$ value. For example, under these assumptions GD(LO) obtains a faster rate than GD(1/L); GD(LO) allows large step sizes that exploit the local smoothness $L(w_t)$ while the iterate-agnostic constant step sizes of GD(1/L) do not take advantage of the local smoothness. But note that different algorithms have different iterates $w_t$ and thus different sequences of local smoothness constants $L(w_t)$. The differing iterate sequences make it difficult to compare two algorithms that both adapt to local smoothness. Thus, under existing local smoothness measures we cannot easily compare the GD(LO) rates depending on $L(w_t)$ to the NAG(1/L) rate which depends on $\sqrt{L}$.

Many algorithms use estimates of local smoothness to speed convergence (Nesterov, 2013; Vainsencher et al., 2015; Schmidt et al., 2017; Fridovich-Keil & Recht, 2019; Liu et al., 2019; Malitsky & Mishchenko, 2020; Park et al., 2021; Patel & Berahas, 2022; Shi et al., 2023; Li & Lan, 2023; Zhang & Hong, 2024). We can use the assumptions of the previous paragraph to show that such algorithms can converge faster than algorithms that do not exploit local smoothness. However, existing tools do not allow us to compare between different algorithms exploiting local smoothness.

A classic argument is that the convergence rate of gradient descent depends only on the Lipschitz constant $L(w_0)$ over the sub-level $\{w \mid f(w) \leq f(w_0)\}$, provided that this sublevel set is closed (Boyd & Vandenberghe, 2004, Chapter 9). Under this argument, we obtain a rate depending on $L(w_0)$ for gradient descent with a suitable constant step size. This assumption can be viewed as assuming glocal smoothness with $\delta$ given by $f(w_0) - f_*$. However, this is a stronger assumption than the glocal assumption in general; glocal smoothness

assumes that a sublevel set with a small $L_*$ exists but does not require the algorithm to be initialized in this set. Further, algorithms can perform better in practice and obtain faster theoretical rates if they can adapt to any value of $\delta$ rather than just $\delta = f(w_0) - f_*$. Indeed, the classic $\delta = f(w_0) - f_*$ assumption suggests GD(LO) performs the same as using a suitable constant step size which does not reflect practical performance.

A line of closely related works are works that assume local smoothness but do not assume global smoothness (Park et al., 2021; Patel & Berahas, 2022; Lu & Mei, 2023; Zhang & Hong, 2024; Malitsky & Mishchenko, 2020). These works tend to focus on achieving global convergence despite the lack of global smoothness. In contrast, our focus is on exploring assumptions under which it is easy to compare the iteration complexities of different algorithms.

The $(L_0, L_1)$ non-uniform smoothness assumption (Zhang et al., 2020) justifies the success of ideas like normalization and gradient clipping when training neural networks. Under this assumption the local smoothness can decrease as the gradient norm decreases. The $(L_0, L_1)$ non-uniform smoothness assumption is thus related to glocal smoothness, as it also allows a smaller smoothness constant near solutions (where the gradient norm is zero). Recent works have shown that gradient descent benefits from non-constant and adaptive step sizes under non-uniform smooothness (Gorbunov et al., 2024) while concurrent work demonstrates the benefits of line search under a variation on non-uniform smoothness (Vaswani & Babanezhad, 2025). However, we have fewer tools to establish $(L_0, L_1)$ non-uniform smoothness than global/local smoothness, and the glocal assumption arguably leads to simpler analyses. Further, in Appendix A we show that a common variant of the $(L_0, L_1)$-smoothness assumption implies glocal smoothness for functions that are either Lipschitz-continuous or Lipschitz-smooth. Thus, in these common Lipschitz settings glocal smoothness is more general and can also arise from other possible assumptions.

Finally, we note that analyses under glocal-smoothness are a special case of regional complexity analysis (Curtis & Robinson, 2021). In this framework we partition the search space of an algorithm into different regions that satisfy different assumptions, and analyze the progress an algorithm makes in each region. Our work considers the simpler case of just two specific regions: the whole space and a sub-level set. We believe that this is an important special case as it leads to simple analyses that still better reflect practical performance. Further, we expect that algorithms that perform well under glocal smoothness should have good performance under other measures of local smoothness.

## 3 Glocal Smoothness

In order to more easily compare optimization algorithms that may exploit local smoothness, we propose a notion of global-local smoothness, which we call *glocal smoothness*:

**Definition 2.** A function $f$ is glocally $(L, L_*, \delta)$-smooth if $f$ is globally $L$-smooth, and locally $L_*$-smooth on the set of $u \in \mathbb{R}^d$ such that $f(u) - f_* \leq \delta$ (where $f_*$ is the infimum of $f$).

Assumptions of this type have been explored for speeding convergence (Vainsencher et al., 2015) and analyzing local optima quality (Mohtashami et al., 2023). Using the notation $\Delta_0 = f(w_0) - f_*$ as the initial sub-optimality, our analyses assume that $\Delta_0 > \delta > \epsilon$ for the desired accuracy $\epsilon$, since without these assumptions there is no need to distinguish between the global and local regions. Our analyses also assume that $f$ has an infimum $f_*$, and some of our analyses will use $w_*$ as a global minimizer of $f$.

### 3.1 Glocal Smoothness of Logistic Regression

Note that $L_* \leq L$ since $L_*$ is measured over a subset of the space, but that for some problems we have $L_* << L$. For example, consider the binary logistic regression loss,

$$\ell(w) := \sum_{i=1}^{n} p(y_i \mid x_i, w),\qquad(3)$$

where $n$ is the number of examples, $x_i \in \mathbb{R}^d$ are the features of example $i$, $y_i \in \{-1, +1\}$ are the labels for example $i$, and $p(y_i \mid x_i, w) = \ln(1 + \exp(-y_i \langle x_i, w \rangle))$. The standard bound on the global smoothness

of binary logistic regression with a data matrix $X$ is $L = (1/4)\|X\|^2$ (Böhning, 1992). The 1/4 factor is the upper bound on $p(y_i = +1 \mid x_i, w)\, p(y_i = -1 \mid x_i, w)$. However, if near solutions we have for all $i$ that $p(y_i = +1 \mid x_i, w) > 0.99$ or $p(y_i = -1 \mid x_i, w) > 0.99$, then $L_*$ is around $(1/100)\|X\|^2$. This allows GD(LO) to eventually take steps that are 25-times larger than those used by GD(1/L) yet still decrease the function. More formally, in Appendix B we show the following result.

**Lemma 3.** The logistic regression loss (3) is glocally $(\frac{1}{4}\|X\|^2, (\ell_* + \delta)\,\|X\|^2, \delta)$-smooth for *all* $\delta > 0$, where $\ell_*$ is the infimum of the loss $\ell$.

For linearly separable data $\ell^* = 0$ and we have glocal $((1/4)\|X\|^2, \delta\|X\|^2, \delta)$-smoothness for all $\delta$. Thus, GD(LO) can take increasing steps that become arbitrarily larger than $1/L$ yet still decrease $\ell$.

### 3.2 Iteration Complexity under Global Smoothness

To illustrate how glocal smoothness allows improved complexities, in this section we first consider strongly convex functions.

**Definition 4.** A function $f$ is $\mu$-strongly convex for some $\mu > 0$ if $\forall u, v \in \mathbb{R}^d$,

$$f(v) \geq f(u) + \langle \nabla f(u), v - u \rangle + \frac{\mu}{2}\|v - u\|^2.$$

Below we review a classic result (Boyd & Vandenberghe, 2004, Chapter 9) regarding the convergence rate of GD(1/L) and GD(LO) on strongly-convex functions.

**Lemma 5.** Assume that $f$ is $\bar{L}$-smooth and $\mu$-strongly convex. For all $t$, for the iterates of gradient descent as defined in (1) with step-size $\eta_t$ given by $1/\bar{L}$ or exact line optimization (2), the guaranteed progress is

$$f(w_t) \leq f(w_{t-1}) - \frac{1}{2\bar{L}}\|\nabla f(w_{t-1})\|^2, \tag{4}$$

which under strong convexity implies

$$f(w_t) - f_* \leq \left(1 - \frac{\mu}{\bar{L}}\right)(f(w_{t-1}) - f_*).$$

This implies that $f(w_t)$ decreases monotonically for all $t$. It also implies that if we start from $\bar{w}_0$ we require

$$T \geq \frac{\bar{L}}{\mu}\log\left(\frac{f(\bar{w}_0) - f_*}{\bar{\epsilon}}\right) \tag{5}$$

iterations to have $f(w_T) - f_* \leq \bar{\epsilon}$. Thus, for a globally $L$-smooth and $\mu$-strongly convex function, the iteration complexity of GD(1/L) and GD(LO) is $(L/\mu)\log(\Delta_0/\epsilon)$.

### 3.3 Iteration Complexity under Glocal Smoothness

The classic iteration complexity (5) under global smoothness is the same for both GD(1/L) and GD(LO). But under strong convexity and *glocal* smoothness, we have the following improved iteration complexity for GD(LO).

**Theorem 6.** Assume that $f$ is glocally $(L, L_*, \delta)$-smooth and $\mu$-strongly convex. For all $t \geq 0$, let $w_t$ be the iterates of gradient descent as defined in (1) with step-size $\eta_t$ given by (2). Then $f(w_T) - f_* \leq \epsilon$ for all

$$T \geq \frac{L}{\mu}\log\left(\frac{\Delta_0}{\delta}\right) + \frac{L_*}{\mu}\log\left(\frac{\delta}{\epsilon}\right).$$

Here we outline the proof, but delay the formal proof until Section 3.5 after addressing a technical issue in Section 3.4. The proof breaks the complexity of the algorithm into two parts. The first part bounds the number of iterations $t_\delta$ required to reach an error of $\delta$ for an $L$-smooth function starting from $w_0$. The

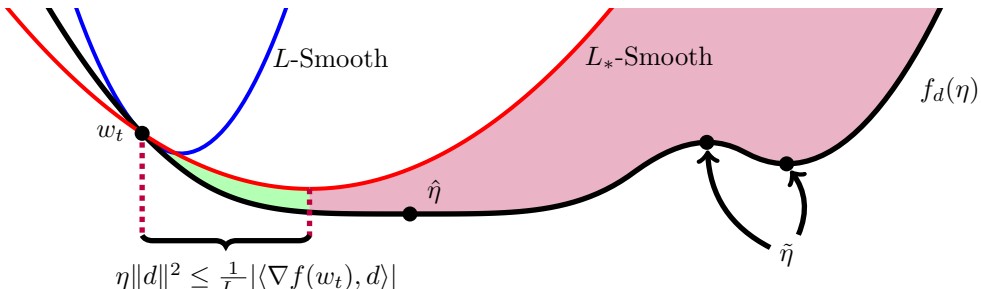

Figure 2: Illustration of corollary 7 along the slice $f_d(\eta) = f(w_t + \eta d)$. While $f_d$ is $L$-smooth globally, it is only $L_*$ smooth over an interval. The bound $\eta\|d\|^2 \leq \frac{1}{L_*}|\langle\nabla f(w_t), d\rangle|$ exactly requires that $\eta$ is smaller than the best step-size under $L_*$-smoothness. Our proof proceeds by arguing that this best step-size cannot exceed the stationary points $\tilde{\eta}$ or the global minimizer $\hat{\eta}$.

second part is the number of iterations to reach an error of $\epsilon$ for an $L_*$-smooth function starting from an iteration $t_\delta$ after which the iterates stay within the local region (recall that we assume $\epsilon < \delta < \Delta_0$). Note that we have omitted the ceiling function $\lceil \cdot \rceil$ from both terms in this (and subsequent) results to present them in a more elegant way (we could alternately add $+1$ to the right side as $T$ must be an integer).

The rate in Theorem 6 for GD(LO) under glocal smoothness is faster than the rate of GD(1/L) whenever $L_* < L$. We note that the GD(LO) rate can also be achieved by using GD(1/L) until we have $f(w_t) - f_* \leq \delta$, and then switching to using GD(1/L$_*$). Indeed, the analysis proceeds by arguing that GD(LO) globally makes at least as much progress as GD(1/L) applied to an $L$-smooth function and locally makes at least as much progress as GD(1/L$_*$) applied to a globally $L_*$-smooth function. However, unlike this GD(1/L)-then-GD(1/L$_*$) hybrid, note that GD(LO) does not need to know any of $L$, $L_*$, or $\delta$. Furthermore, GD(LO) simultaneously obtains the fast rate for all values of $\delta$ (and their corresponding $L_*$ values).

### 3.4 Fast Local Progress of Descent Methods

After first reaching the region of $L_*$-smoothness, the proof of Theorem 6 relies on bounding the function decrease achieved by GD(LO) with the amount achieved by GD(1/L$_*$) on a globally $L_*$-smooth function. However, a potential challenge is that a step size of $1/L_*$ may be too large to stay in the region of $L_*$ smoothness. Indeed, for this reason classic arguments for fast convergence on sublevel sets either require the sublevel set to be closed (Boyd & Vandenberghe, 2004, Chapter 9) or require that the sublevel set is bounded and use an $L_*$ over a bigger set than the sublevel set (Bertsekas, 1997, Exercise 1.2.5). However, in Appendix C we show the following result which implies that a step size of $1/L_*$ cannot leave the $L_*$-smooth region even if the sublevel set is open and even if the objective is non-convex (see fig. 2 for a visualization).

**Proposition 7.** Assume that $f$ is $(L, L_*, \delta)$ glocally smooth, $f(w_t) - f_* \leq \delta$, and $d$ is a descent direction for $f$ at $w_t$, meaning that $\langle\nabla f(w_t), d\rangle < 0$. For any step-size $\eta_*$ s.t. $\eta_*\|d\|^2 \leq \frac{1}{L_*}|\langle\nabla f(w_t), d\rangle|$, $f(w_t + \eta_* d) - f_* \leq \delta$.

We will use this result to prove convergence for various algorithms under different assumptions, as it holds for any algorithm that uses a descent direction. The next corollary is a specialization of Proposition 7 that bounds the function decrease of GD(LO) within the sublevel set (see Appendix C for the proof).

**Corollary 8.** Assume that $f$ is $(L, L_*, \delta)$ glocally smooth. Additionally, assume that $w_t$ satisfies $f(w_t) - f_* \leq \delta$. Then for gradient descent with steps defined in (1) with step size $\frac{1}{L_*}$, $f(w_{t+1}) - f_* \leq \delta$. In addition, the guaranteed progress is at least $f(w_{t+1}) \leq f(w_t) - \frac{1}{2L_*}\|\nabla f(w_t)\|^2$ if instead (2) is used to select the step size.

### 3.5 Outline of Two-Phase Proof Strategies and Proof of Theorem 6

The argument used to formally show Theorem 6 follows a two-phase structure where we first bound the complexity to reach and stay in a local region, and then bound the complexity within the local region. We note that such two-phase proofs are common in other settings such as the classic analysis of Newton's

method (Boyd & Vandenberghe, 2004, see Chapter 9) and analyses of gradient descent under $(L_0, L_1)$ non-uniform smoothness. Many of our analyses will also follow two-phase structures. Thus, rather than repeating all logical steps explicitly, we will show that certain sufficient ingredients lead to a corresponding iteration complexity. In particular, consider the following procedure:

1. Define a "region of interest".

2. For an iterate $t_\delta$, give a condition under which all iterates $w_t$ for $t \geq t_\delta$ are guaranteed to stay within the region of interest.

3. Based on global properties of the function and beginning from $w_0$, upper bound the number of iterations to reach such an iteration $t_\delta$.

4. Based on local properties of the region of interest and beginning from $w_{t_\delta}$, upper bound the number of iterations to reach an accuracy $\epsilon$.

5. Use the properties of $t_\delta$ and the region of interest to remove the dependence on the particular iterate $w_{t_\delta}$.

Given the above steps, it follows that the overall iteration complexity is the sum of the bounds in points 3 and 5. We now follow this recipe to give the proof of Theorem 6.

*Proof of Theorem 6.* By following the analysis structure outlined above we have:

1. Our region of interest under glocal smoothness is the set $\{w \mid f(w) - f_* \leq \delta\}$.

2. Let $t_\delta$ be the first iteration such that $f(w_{t_\delta}) - f_* \leq \delta$. By monotonicity of $f(w_t)$ in terms of $t$ from (4), we have that $f(w_t) - f_* \leq \delta$ for all $t \geq t_\delta$ so the iterates stay within the region.

3. Initialized at $w_0$, using Lemma 5 with $\bar{L} = L$ and $\bar{\epsilon} = \delta$, GD(LO) is guaranteed to satisfy $f(w_t) - f_* \leq \delta$ after $t_\delta = \left\lceil \frac{L}{\mu} \log \left( \frac{f(w_0) - f_*}{\delta} \right) \right\rceil$ iterations.

4. Initialized at $w_{t_\delta}$, by Corollary 8 the guaranteed progress of using an exact line search for $t \geq t_\delta$ is at least that of using a step size of $\frac{1}{L_*}$ on a globally $L_*$-smooth function. Thus we can use Lemma 5 with $\bar{L} = L_*$ and $\bar{\epsilon} = \epsilon$ to obtain that GD(LO) starting from $\bar{w}_0 = w_{t_\delta}$ is guaranteed to have $f(w_t) - f_* \leq \epsilon$ after another $t = \left\lceil \frac{L_*}{\mu} \log \left( \frac{f(w_{t_\delta}) - f_*}{\epsilon} \right) \right\rceil$ iterations.

5. Using that $f(w_{t_\delta}) - f_* \leq \delta$, we can conclude that it is sufficient to have $t = \left\lceil \frac{L_*}{\mu} \log \left( \frac{\delta}{\epsilon} \right) \right\rceil$ additional iterations after $t_\delta$.

$\square$

### 3.6 Comparing Line Search to Acceleration

A key advantage of the glocal assumption is that it allows us to compare algorithms that address local smoothness in different ways. In particular, ignoring constant factors we note that the rate of GD(LO) under glocal smoothness and strong convexity is faster than NAG(1/L) if

$$\frac{L_*}{L} \ll \frac{(\sqrt{\kappa})^{-1} \log(\Delta_0/\epsilon) - \log(\Delta_0/\delta)}{\log(\delta/\epsilon)}. \tag{6}$$

This is possible if $L_* < L$, the condition number $\kappa = \frac{L}{\mu}$ is not too large, and $\delta$ is not too small. In words, we expect line search to outperform acceleration if the problem is not too badly conditioned and there is a large region around the minimizer with a smaller local smoothness constant than the global smoothness constant.

It is interesting to consider how the limiting behaviour of (6) agrees with the classic results and our intuitions. For example, NAG(1/L) is faster than GD(LO) for sufficiently large condition numbers. However, GD(LO) outperforms NAG(1/L) for any sufficiently small condition number (assuming $L_* < L$ and $\delta < \Delta_0$). As another example, as $\delta$ approaches $\Delta_0$ the algorithm spends more time in the local region and we have that GD(LO) outperforms NAG(1/L) if $L_* < \sqrt{L\mu}$. But as $\delta$ approaches $\epsilon$ we spend less time in the local region and NAG(1/L) is the preferred method.

### 3.7 Iteration Complexity of Logistic Regression under Glocal Smoothness

We state Theorem 6 in terms of a specific value of $\delta$, but this bound holds for all valid values of $\delta$. Thus, in cases where we know how $L_*$ depends on $\delta$, we can obtain the tightest bound by minimizing across $\delta$ values. In this section, we minimize over $\delta$ for logistic regression using the bound in Section 3.1.

We first write out the classic iteration complexity bound (5) for GD(1/L) and GD(LO) on logistic regression based on the global smoothness bound $L \leq (1/4)\|X\|^2$,

$$T \geq \frac{\|X\|^2}{\mu}\left(\frac{1}{4}\log\left(\frac{\Delta_0}{\epsilon}\right)\right). \tag{7}$$

We now give the result obtained under the glocal smoothness bound in Lemma 3 by minimizing over $\delta$ in Theorem 6 (see Appendix B) subject to $\Delta_0 \geq \delta \geq \epsilon$.

**Proposition 9.** Assume that $f$ is the logistic regression loss (3) and that $f$ is $\mu$-strongly convex. For all $t \geq 0$, let $w_t$ be the iterates of gradient descent as defined in (1) with step-size $\eta_t$ given by (2). Then, depending on the value of $\xi = 1/4 - \ell_*$ we have $f(w_T) - f_* \leq \epsilon$ for all

$$\text{Case 1:} \quad T \geq \frac{\|X\|^2}{\mu}\left(\frac{1}{4}\log\left(\frac{\Delta_0}{\epsilon}\right)\right) \qquad\qquad \text{if } \xi \leq \epsilon,$$

$$\text{Case 2:} \quad T \geq \frac{\|X\|^2}{\mu}\left((\ell_* + \Delta_0)\log\left(\frac{\Delta_0}{\epsilon}\right)\right) \qquad\qquad \text{if } \xi \geq \Delta_0\log(\Delta_0 e/\epsilon),$$

$$\text{Case 3:} \quad T \geq \frac{\|X\|^2}{\mu}\left(\frac{1}{4}\log\left(\frac{\Delta_0}{\epsilon}\right) - \xi\frac{(\omega-1)^2}{\omega}\right) \qquad\qquad \text{otherwise.}$$

where $\omega$ is the value of the Lambert $W$ function evaluated at the value $\xi e/\epsilon$ (which is unique because $\xi e/\epsilon$ is positive in Case 3).

We can think of $\xi$ as the amount that the local smoothness bound improves over the global smoothness bound. In the first case the improvement $\xi$ is negative or is so small that the optimal value of $\delta$ is $\epsilon$ and there is no improvement over the global smoothness bound. In the second case the improvement $\xi$ is so large that the optimal value of $\delta$ is $\Delta_0$ and in this case we use the local bound starting from the first iteration. We note that the iteration complexity in the second case is smaller than the bound in the first case (when the condition holds). In the third case the optimal value of $\delta$ is between $\epsilon$ and $\Delta_0$, and the (negative) second term is the improvement obtained over the global bound. We note that both Cases 2 and 3 give a strictly-improved iteration complexity bound for GD(LO) over the standard result (7), and that Case 3 continuously interpolates between Case 1 (which reverts to using the global bound) and Case 2 (where we define the local region based on the starting iterate).

## 4 Practical Algorithms for Step Size Selection

We can efficiently perform LO numerically for some problems like linear models and 2-layer neural networks (Shea & Schmidt, 2024). However, neither GD(1/L) or GD(LO) are practical algorithms for many problems as it may not be possible in practice to compute the Lipschitz constant $L$ or perform LO efficiently. In this section we discuss practical methods to set the step size for general problems, and consider their iteration complexities under the glocal assumption.

Under global smoothness, a backtracking line search procedure (Beck & Teboulle, 2009) can achieve the rate of GD(1/L). This method starts with an estimate $L_0$ of $L$, initializes $L_t = L_{t-1}$ for $t > 0$, and doubles the

value of $L_t$ whenever the Armijo sufficient decrease condition (Armijo, 1966)

$$f(w_t) \leq f(w_{t-1}) - \alpha \eta_t \|\nabla f(w_{t-1})\|^2, \tag{8}$$

is not satisfied with $\eta_t = 1/L_t$ and $\alpha = 1/2$. If $L_0 \leq L$, this procedure achieves the GD(1/L) rate of $O((L/\mu) \log(\Delta_0/\epsilon))$. Unfortunately, backtracking does not achieve a faster rate under glocal smoothness since it never decreases its guess $L_t$ of $L$ (so it does not increase the step size $\eta_t$ in the local region).

It is possible to obtain an improved iteration complexity over GD(1/L) under glocal smoothness in practice using a backtracking procedure with resets (Scheinberg et al., 2014). When the guess $L_t$ is reset to $L_0$ on each iteration, if $L_0 \leq L$ the rate of gradient descent is improved to $O((L/\mu) \log(\Delta_0/\delta) + (\max\{L_*, L_0\}/\mu) \log(\delta/\epsilon))$ under the glocal assumption. This backtracking-with-resets method achieves the fast rate of GD(LO) if $L_0 \leq L_*$. However, this method may require significant backtracking on each iteration. Backtracking with resets also ideally requires knowledge of $L_*$ as it has a worse complexity than GD(LO) if $L_0 > L_*$. Common variations to reduce the amount of backtracking are to use a line search that does not require monotonicity (Grippo et al., 1986; Raydan, 1997) or to propose more clever increases in the estimate of $L$ (Cavalcanti et al., 2024).

We can achieve the rate of GD(LO) under glocal smoothness using a procedure that includes both a forwardtracking and a backtracking step (Fridovich-Keil & Recht, 2019). We present the following formal convergence result for this method as follows.

**Theorem 10.** Assume that $f$ is glocally $(L, L_*, \delta)$-smooth and $\mu$-strongly convex. Assume $0 < \alpha \leq 1/2$ and $0 < \beta < 1$. For all $t \geq 0$, let $w_t$ be the iterates of gradient descent as defined in (1) with step-size $\eta_t$ satisfying (8) with a forwardtracking-backtracking procedure. Then $f(w_T) - f_* \leq \epsilon$ for all

$$T \geq \frac{L}{2\beta\alpha\mu} \log\left(\frac{\Delta_0}{\delta}\right) + \frac{L_*}{2\beta\alpha\mu} \log\left(\frac{\delta}{\epsilon}\right).$$

We give the proof of this theorem as well as more details about the forwardtracking step in Appendix D. Notice that the procedure nearly achieves the rate of GD(LO) in Theorem 6, and matches the GD(LO) rate as $\beta$ approaches 1 and $\alpha$ approaches $1/2$. Unfortunately, the iterations of the method above can be expensive since the step size could be increased or decreased many times within each iteration of the algorithm. However, many practical heuristics exist to reduce the cost of such algorithms (Moré & Thuente, 1994; Nocedal & Wright, 1999).

We note that it is also possible to achieve a rate similar to GD(LO) under glocal smoothness without increasing the iteration cost if we know $f_*$ and use the Polyak step size (Polyak, 1987),

$$\eta_t = \frac{f(w_t) - f_*}{\|\nabla f(w_t)\|^2}. \tag{9}$$

**Theorem 11.** Assume that $f$ is glocally $(L, L_*, \delta)$-smooth and $\mu$-strongly convex. For all $t \geq 0$, let $w_t$ be the iterates of gradient descent as defined in (1) with step-size $\eta_t$ given by (9). Then $f(w_T) - f_* \leq \epsilon$ for all

$$T \geq 4\left(\frac{L}{\mu} \log\left(\frac{L}{2} \frac{\|w_0 - w_*\|^2}{\delta}\right) + \frac{L_*}{\mu} \log\left(\frac{L_* \delta}{L \epsilon}\right)\right).$$

See Appendix E.1 for the proof of this result. This result is similar to GD(LO), with a worse dependence inside the first logarithmic factor (but a better dependence inside the second logarithmic factor). This is because the Polyak step size does not guarantee that the loss decreases monotonically. Thus, the first term must bound the time until the algorithm will never leave the $\delta$ region. We give a simple example where the Polyak step size increases the function in Appendix E.2. In the next section, we show that a similar rate can be achieved without knowing $f^*$ using the AdGD step size (Malitsky & Mishchenko, 2020).

## 5 Other Assumptions

The previous sections considered globally strongly-convex functions and glocal smoothness based on sublevel sets. In this section, we first define glocal smoothness based on the distance to the solution. We then consider alternatives to global strong-convexity.

## 5.1 Iterates Version of Glocal Smoothness

We previously defined glocal smoothness in terms of function values (Definition 2). However, some choices of step sizes like the Polyak step size guarantee a decrease in the distance of the iterate to the solution on each iteration but do not guarantee a decrease in the function value on each iteration. To highlight the advantage of algorithms whose progress is best measured in iterate distance, it is more natural to define glocal smoothness in terms of the iterate distance to the solution (rather than the optimality gap).

**Definition 12.** A function $f$ is glocally $(L, L_*, \delta)$-smooth in iterates if $f$ is globally $L$-smooth, and locally $L_*$-smooth for all $w \in \mathbb{R}^d$ such that $\|w - w_p\|^2 \leq \delta$ for some $\delta > 0$ where $w_p$ is the projection of $w$ onto the solution set.

In the case of strongly-convex functions, there is a unique solution and thus $w_p = w_*$ for all $w$. Similar to before, we will assume $\|w_0 - (w_0)_p\|^2 > \delta > \epsilon > 0$. For the remainder of this paper, we will state that $f$ is glocally smooth in iterates when we use this alternate definition. Under this modified version of the glocal smoothness assumption, we can obtain a simpler iteration complexity (see Appendix E.3) of gradient descent with the Polyak step size. In particular, since the Polyak step size leads to monotonic decrease in the distance to the solution, by assuming glocal smoothness in the iterates we can remove the factors of $L$ and $L_*$ inside the logarithms in Theorem 11.

We now consider the AdGD step size (Malitsky & Mishchenko, 2020) for gradient descent in the context of glocal smoothness. The AdGD step size takes the following form: at iteration $t$, we set $\theta_{t-1} = \frac{\eta_{t-1}}{\eta_{t-2}}$ and,

$$\eta_t = \min\left( \sqrt{1 + \frac{\theta_{t-1}}{2}}\, \eta_{t-1}, \frac{w_t - w_{t-1}}{2\|\nabla f(w_t) - \nabla f(w_{t-1})\|} \right). \tag{10}$$

In Appendix F, we show that the AdGD step size under glocal smoothness achieves a similar rate to GD(LO) (without LO) and to the Polyak step size (without knowledge of $f_*$) beginning at the third iteration.

**Theorem 13.** Assume that $f$ is glocally $(L, L_*, \delta)$-smooth in iterates and $\mu$-strongly convex. For all $t > 2$, let $w_t$ be the iterates of gradient descent as defined in (1) with step-size $\eta_t$ given by (10). Then $\|w_T - w_*\|^2 \leq \epsilon$ for all

$$T \geq O\left( \frac{L}{\mu} \log\left( \frac{\Phi_3}{\delta} \right) + \frac{L_*}{\mu} \log\left( \frac{\delta}{\epsilon} \right) \right),$$

where $\Phi_3 = \|w_3 - w_*\|^2 + \frac{1}{2}(1 + \frac{2\mu}{L})\|w_3 - w_2\|^2 + 2\eta_2(1 + \theta_2)(f(w_2) - f_*)$.

## 5.2 Glocal Strong Convexity

Similar to glocal smoothness (Definition 2), one could make a similar assumption that the function $f$ is globally $\mu$-strongly convex and locally $\mu_*$-strongly convex when $f(x) - f_* \leq \delta$ for some $\delta > 0$ and $\mu_* > 0$. If $f$ glocally smooth and glocally strong convex, the iteration complexity for GD(LO) is improved to

$$T \geq \frac{L}{\mu} \log\left( \frac{\Delta_0}{\delta} \right) + \frac{L_*}{\mu_*} \log\left( \frac{\delta}{\epsilon} \right). \tag{11}$$

This is the complexity given in Theorem 6 with $\mu$ in the second term replaced with $\mu_*$, and is an improved rate when $\mu_* > \mu$ (while $\mu_* \geq \mu$ by definition). An example where we can have $\mu_* > \mu$ is the Huber loss for robust regression (Hastie et al., 2017). Note that GD(1/L) is also able to adapt to a local $\mu_*$ even though it does not adapt to $L_*$, obtaining (11) but with $L_*$ replaced by $L$.

## 5.3 Globally Convex Case

A variation on the assumption of the previous section is functions that are locally $\mu_*$-strongly convex, but only convex globally (corresponding to the degenerate case of $\mu = 0$). To analyze this case we define (Nesterov, 2012; Beck & Tetruashvili, 2013)

$$R^2(\rho) = \max_{w \in \mathbb{R}^d, w_* \in W_*} \{\|w - w_*\|^2 : f(w) \leq \rho\},$$

for a given $\rho \in \mathbb{R}$. Note that $W_*$ refers to the set of minimizers of $f$, which under local strong-convexity only contains the unique solution $w_*$. In this setting, GD(1/L) has a sublinear convergence rate leading to an $O((L/\epsilon)\|x_0 - x_*\|^2)$ complexity, while GD(LO) has the following complexity (Appendix G).

**Theorem 14.** Assume that $f$ is glocally $(L, L_*, \delta)$-smooth and locally $\mu_*$-strongly convex. Additionally, assume that $f$ is globally convex and coercive. For all $t \geq 0$, let $w_t$ be the iterates of gradient descent as defined in (1) with step-size $\eta_t$ given by (2). Then $f(w_T) - f_* \leq \epsilon$ for all

$$T \geq O\left(\frac{L}{\delta}R^2(f(w_0)) + \frac{L_*}{\mu_*}\log\left(\frac{\delta}{\epsilon}\right)\right).$$

In this setting GD(LO) initially has a sublinear rate, with a worse constant of $R^2(f(w_0))$ compared to $\|x_0 - x_*\|^2$ for GD(1/L). Within the local region GD(1/L) and GD(LO) both obtain a linear convergence rate, but GD(LO) obtains a faster rate based on $L_*$ instead of $L$. It is unclear if the worse constant in the sublinear part is necessary for GD(LO), but for sufficiently large $\delta$ and/or small $\epsilon$ the GD(LO) complexity improves over the GD(1/L) complexity. We note that the iterates do not necessarily converge for GD(LO) on convex functions (Bolte & Pauwels, 2022), but the function values are guaranteed to decrease and the iterates subsequently converge under local strong convexity. In Appendix G we also consider functions that are globally convex and only locally convex, rather than locally strongly convex. Note that unlike the case when $f$ is locally strongly convex, the solution set $W_*$ may contain multiple minima if $f$ only satisfies local convexity.

### 5.4 PL Condition

An alternative relaxation of strong convexity is the PL inequality (Polyak, 1963; Karimi et al., 2016). The PL inequality holds if for some $\mu > 0$ it holds for all $u$ that

$$\frac{1}{2}\|\nabla f(u)\|^2 \geq \mu(f(u) - f_*). \tag{12}$$

Note that Proposition 7 does not require convexity and thus holds for PL functions. Indeed, strong convexity is only used to obtain the PL inequality in Theorem 6. Thus, for PL functions the iteration complexity of GD(LO) is the same as that given in Theorem 6 (with $\mu$ being the PL constant). Similar to the locally-convex case, we highlight that this improved complexity under the PL inequality is possible despite the fact that PL functions can have multiple minimizers.

It is worth noting that if a function $f$ is $\mu$-strongly convex, then $f$ is also $\mu$-PL. But a function could satisfy the PL condition and not be convex. Although the PL inequality is too strong to hold globally for neural networks, the PL inequality has been proposed as a reasonable local model of several neural network loss landscapes that explains the fast convergence of gradient-based methods. For example, Liu et al. (2022) show sufficiently-wide over-parameterized networks satisfy the PL inequality around their initialization point. More recently, Aich et al. (2025) show that a local variant of the PL inequality holds in locally quasi-convex regions satisfying a local neural tangent kernel (NTK) stability assumption.

## 6 Other Algorithms under Glocal Smoothness

Up to this point we have focused on gradient descent under various step sizes. We now consider showing how other algorithms can benefit from variants of glocal smoothness. We consider in particular random and greedy coordinate descent, stochastic gradient in the interpolation setting, and NAG. In Appendix K, we also consider the non-linear conjugate gradient method when the Hessian is well-behaved in the local region.

### 6.1 Coordinate Descent

For each step $t$, coordinate descent takes steps of the form

$$w_{t+1} = w_t - \eta_t \nabla_{j_t} f(w_t) e_{j_t}, \tag{13}$$

for some positive step size $\eta_t$. Here, $\nabla_j f(w)$ is the $j$th coordinate of $\nabla f(w)$ and $e_{j_t}$ refers to the basis vector with entry 1 at index $j_t$ and all other entries 0. Coordinate descent and its variations (Nesterov, 2012; Nutini et al., 2015; Wright, 2015; Locatello et al., 2018) have been explored extensively. The analysis of coordinate descent typically uses coordinate-wise variations on smoothness.

**Definition 15.** A function $f$ is coordinate-wise $L$-smooth if $\forall u \in \mathbb{R}^d$, we have that for each $j \in [d]$ and any $\alpha \in \mathbb{R}$,

$$|\nabla_j f(u + \alpha e_j) - \nabla_j f(u)| \leq L|\alpha|.$$

We consider a corresponding coordinate-wise version of glocal smoothness.

**Definition 16.** A function $f$ is coordinate-wise glocally $(L, L_*, \delta)$-smooth if $f$ is globally coordinate-wise $L$-smooth, and locally coordinate-wise $L_*$-smooth for all $u \in \mathbb{R}^d$ such that $f(u) - f_* \leq \delta$.

Line optimization is also a natural choice for coordinate descent, albeit with a restriction to single-coordinate updates,

$$\eta_t \in \arg\min_{\eta} f(w_t - \eta \nabla_{j_t} f(w_t) e_{j_t}). \tag{14}$$

We first consider randomized coordinate descent where $j_t$ is selected such that each coordinate $j$ is equally probable. Under this uniform random sampling strategy, the iteration complexity for coordinate descent with a step size of $1/L$ under the standard global smoothness assumption is $\frac{dL}{\mu} \log(\frac{\Delta_0}{\epsilon})$ (Nesterov, 2012). Under coordinate-wise glocal smoothness, we obtain the following improved iteration complexity with high probability using LO (see Appendix H).

**Theorem 17.** Assume that $f$ is coordinate-wise glocally $(L, L_*, \delta)$-smooth and $\mu$-strongly convex. For all $t \geq 0$, let $w_t$ be the iterates of coordinate descent as defined in (13) with $j_t$ selected from the uniform distribution over $\{1, 2, \ldots, d\}$ and step-size $\eta_t$ given by (14). Define $X_\delta$ as the event that $f(w_{t_\delta}) - f_* \leq \delta$, where $t_\delta = \lceil \frac{dL}{\mu} \log\left(\frac{\Delta_0}{\delta\zeta}\right) \rceil$. Then $E[f(w_T) \mid X_\delta] - f_* \leq \epsilon$ for all

$$T \geq \frac{dL}{\mu} \log\left(\frac{\Delta_0}{\delta\zeta}\right) + \frac{dL_*}{\mu} \log\left(\frac{\delta}{\epsilon}\right),$$

with probability $1 - \zeta$, where $d$ is the problem dimension and $E[f(w_T) \mid X_\delta]$ denotes the expectation over the randomness in the coordinate selection, conditioned on the event $X_\delta$.

A challenge in analyzing randomized coordinate descent is that it is not deterministically guaranteed that the iterates of coordinate descent will enter the $\delta$ sublevel set by a given iteration. Our analysis thus conditions on this happening with high probability after a relevant number of iterations. Note that we only need to condition on the algorithm reaching the sublevel set at a single iteration, since the algorithm subsequently stays within the sublevel set by the monotonic decrease in function values.

We now consider the canonical greedy coordinate descent method, where we set $j_t \in \arg\max_j |\nabla_j f(w_t)|$ (Nutini et al., 2015). In Appendix H, using strong convexity as measured in the 1-norm, we show the following improved iteration complexity under glocal smoothness when selecting the coordinate greedily.

**Theorem 18.** Assume that $f$ is coordinate-wise glocally $(L, L_*, \delta)$-smooth and $\mu_1$-strongly convex in the 1-norm. For all $t \geq 0$, let $w_t$ be the iterates of coordinate descent as defined in (13) with the greedy rule $j_t \in \arg\max_j |\nabla_j f(w_t)|$ and step-size $\eta_t$ given by (14). Then $f(w_T) - f_* \leq \epsilon$ for all

$$T \geq \frac{L}{\mu_1} \log\left(\frac{\Delta_0}{\delta}\right) + \frac{L_*}{\mu_1} \log\left(\frac{\delta}{\epsilon}\right).$$

No high probability bound is needed here as the algorithm is deterministic.

## 6.2 Stochastic Gradient Descent

In this section, we consider the stochastic gradient descent (SGD) algorithm (Robbins & Monro, 1951). For simplicity, we will consider functions that can be written as a finite sum, $f(w) = \sum_{i=1}^{n} f_i(w)$. On iteration $t$ SGD takes steps of the form

$$w_{t+1} = w_t - \eta_t \nabla f_{i_t}(w_t), \tag{15}$$

where $i_t$ is sampled with equal probability from $\{1, \ldots n\}$. An additional assumption that we use in our analysis is interpolation, which holds for many modern over-parameterized models (Zhang et al., 2016; Ma et al., 2018).

**Definition 19.** A function $f$ satisfies interpolation if $\nabla f(w_*) = 0$ implies that $\nabla f_i(w_*) = 0$ for $i \in [n]$.

That is, if $w_*$ is a stationary point of $f$, then $w_*$ is also a stationary point of each individual function $f_i$ for all $i \in [n]$. However, note that we do not assume the individual $f_i$ are strongly-convex and the individual $f_i$ may have different sets of stationary points. We analyze a variant on the stochastic Armijo rule (Vaswani et al., 2019) for step size selection, where on each iteration we initialize the step size $\eta$ to some value $\eta_{\max}$ and divide it in half until we have

$$f_{i_t}(w_{t+1}) \leq f_{i_t}(w_t) - \frac{1}{2\eta}\|\nabla f_{i_t}(w_t)\|^2. \tag{16}$$

We give the following iteration complexity for SGD. See Appendix I for the proof.

**Theorem 20.** Assume that $f$ is $\mu$-strongly convex and that each $f_i$ is convex and glocally $(L_{\max}, L_{\max,*}, \delta)$-smooth in iterates. Furthermore, assume that $f$ satisfies the interpolation assumption given in Definition 19. For all $t \geq 0$, let $w_t$ be the iterates of stochastic gradient descent as defined in (15) with step-size $\eta_t$ given by backtracking from some $\eta_{\max} \geq 1/L_{\max,*}$ by dividing the step size in half until we satisfy (16). Define $X_\delta$ as the event that $\|w_{t_\delta} - w_*\| \leq \delta$, where $t_\delta = \left\lceil \frac{2L_{\max}}{\mu} \log \left( \frac{\|w_0 - w_*\|^2}{\delta\zeta} \right) \right\rceil$. Then $E[\|w_T - w_*\| \mid X_\delta] \leq \epsilon$ for all

$$T \geq 2 \left( \frac{L_{\max}}{\mu} \log \left( \frac{\|w_0 - w_*\|^2}{\delta\zeta} \right) + \frac{L_{\max,*}}{\mu} \log \left( \frac{\delta}{\epsilon} \right) \right).$$

with probability $1 - \zeta$, where $E[\|w_T - w_*\| \mid X_\delta]$ is a random variable through $X_\delta$.

A key property that allows us to show this result is that the interpolation assumption implies that the iterates will deterministically move closer to the minimizer on each step (for a sufficiently small constant step size). Similar to the coordinate descent case, we use that the iterates reach the local region after a certain number of iterations with high probability and then stay within this region.

### 6.3 Nesterov's Accelerated Gradient Method

In this section, we show that glocal smoothness can also lead to improved rates for Nesterov's accelerated gradient method (NAG) when combined with an Armijo line search. We present our results for the three-sequence variant of NAG, which has the following update,

$$\begin{aligned} y_t &= w_t + \frac{\sqrt{q_t}}{1 + \sqrt{q_t}} (z_t - w_t) \\ w_{t+1} &= y_t - \eta_t \nabla f(y_t) \\ z_{t+1} &= (1 - \sqrt{q_t})z_t + \sqrt{q_t}(y_t - \frac{1}{\mu}\nabla f(y_t)), \end{aligned} \tag{17}$$

where $q_t = \eta_t \cdot \mu$ and the step-size $\eta_t$ is computed using backtracking line-search on a two-step Armijo condition (see (Nesterov et al., 2018, Eq. 2.2.7)),

$$\begin{aligned} y_t &= w_t + \frac{\sqrt{q_t}}{1 + \sqrt{q_t}} (z_t - w_t) \\ f(w_{t+1}) &\leq f(y_t) - \frac{\eta_t}{2}\|\nabla f(y_t)\|_2^2. \end{aligned} \tag{18}$$

Notice that the line search is performed through the computation of $y_t$, rather than just on $w_t$. We prove in corollary 36 that eq. (17) is formally equivalent to the standard momentum formulation of NAG (despite the use of a variable step size). Now we are ready to give the iteration complexity for NAG with backtracking line search under the glocal smoothness assumption.

**Theorem 21.** Assume that $f$ is glocally $(L, L_*, \delta)$-smooth and $\mu$-strongly convex. Let $w_t$ be the iterates of NAG as defined in (17) with step-size $\eta_t$ given by backtracking from some $\eta_{\max} \geq 1/L_*$ by dividing the step size in half until we satisfy (18). Then $f(w_T) - f_* \leq \epsilon$ for all

$$T \geq \sqrt{\frac{2L}{\mu}} \log\left(\frac{L}{\mu} \frac{\left(\Delta_0 + \frac{\mu}{2}\|w_0 - w_*\|^2\right)}{\delta}\right) + \sqrt{\frac{2L_*}{\mu}} \log\left(\frac{\mu}{L}\frac{\delta}{\epsilon}\right).$$

See Appendix J for the proof, which extends the potential function approach of previous work (d'Aspremont et al., 2021). Compared to gradient descent (Theorem 6), NAG with Armijo line search must converge to the sub-optimality $\tilde{\delta} = L\delta/\mu$ before the complexity improves due to local $L_*$-smoothness. This is because the extrapolation sequence $y_t$ converges $O(L/\mu)$ slower than $w_t$, but must also enter the $\delta$ sub-optimal set before the line-search can provably benefit from the improved smoothness.

## 7 Conclusion

Compared to popular global smoothness assumptions, we believe that worst-case bounds under glocal smoothness better reflect the performance of numerical optimization algorithms in practice than classic global bounds. Further, a key advantage of glocal smoothness compared to many previous local smoothness assumptions is that glocal smoothness allows comparisons between algorithms. Indeed, we have given a precise condition under which "line search can really help" in the sense that using line search with the basic GD method can lead to a significantly improved iteration complexity bound than using acceleration without line search.

We have shown that glocal assumptions lead to improved iteration complexities in a wide variety of settings. Nevertheless, there remains a huge variety of open problems related to the glocal assumption. We highlight many open problems in Appendix M, not only related to optimization algorithms but also the relevance of glocal assumptions to neural networks. We encourage theoreticians to consider these open problems as they may not only lead to better iteration complexities but also to algorithms that perform better in practice.

**Acknowledgments**

We thank Betty Shea and Reza Babanezhad for providing valuable insights on this project and Adrien Taylor for their detailed correspondence and support regarding the accelerated results. We also thank Lieven Vandenberghe for clarifications regarding Section 3.4, Stephen Vavasis for highlighting the local properties of the Huber loss, and Katya Scheinberg for suggesting the glocal name. The work was partially supported by the Canada CIFAR AI Chair Program and NSERC Discovery Grant RGPIN-2022-036669. Aaron Mishkin was supported by NSF Grant DGE-1656518 and by NSERC Grant PGSD3-547242-2020.

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

## Appendix Table of Contents

## A  Connection to $(L_0, L_1)$ Non-Uniform Smoothness

We focus on a specific variant of the $(L_0, L_1)$ non-uniform smoothness assumption (Chen et al., 2023), and show that it implies the glocal smoothness assumption if the function under consideration is either $G$-Lipschitz or $L$-smooth.

**Definition 22.** ( (Chen et al., 2023, Definition 2.1)) A function $f$ is $(L_0, L_1)$ non-uniform asymmetric smooth if $\forall x, y$,

$$\|\nabla f(x) - \nabla f(y)\| \leq (L_0 + L_1 \|\nabla f(x)\|) \|x - y\|.$$

The above assumption does not require twice-differentiability, unlike the non-uniform smoothness assumption given in prior work (Zhang et al., 2020). However, if $f$ is also twice differentiable, Definition 22 is stronger than this definition (refer to (Chen et al., 2023, Theorem 1)).

**Proposition 23.** Assume that $f$ satisfies $(L_0, L_1)$ non-uniform smoothness. Then for any $\delta$,

1. If $f$ is $G$-Lipschitz, then $f$ is $(L_0 + L_1 G, L_0 + L_1(4L_1\delta + 2\sqrt{L_0\delta}), \delta)$-glocal smooth.

2. If $f$ is $L$-smooth, then $f$ is $(L, L_0 + L_1(4L_1\delta + 2\sqrt{L_0\delta}), \delta)$-glocal smooth.

*Proof.* Recall that a function is $G$-Lipschitz if $\|\nabla f(x)\| \leq G$ for all $x$. If $f$ also satisfies $(L_0, L_1)$ non-uniform smoothness, then

$$\|\nabla f(x) - \nabla f(y)\| \leq (L_0 + L_1 G) \|y - x\|.$$

Hence, $f$ is globally $(L_0 + L_1 G)$-smooth. If instead $f$ is $L$-smooth, then by definition it is globally $L$-smooth.

We now proceed to show that $f$ is locally $L_0 + L_1(4L_1\delta + 2\sqrt{L_0\delta})$ smooth. Using existing results (Chen et al., 2023, Theorem 1, Proposition 3.2), we know that if $f$ is $(L_0, L_1)$ non-uniform smooth, then the following descent lemma holds for all $x, y$,

$$f(y) \leq f(x) + \langle \nabla f(x), y - x \rangle + \left( \frac{L_0 + L_1\|\nabla f(x)\|}{2} \right) \exp(L_1\|x - y\|)\|y - x\|^2. \tag{19}$$

Following a proof similar to previous work (Gorbunov et al., 2024, Lemma 2.2), we set $y = x - \frac{\nu \nabla f(x)}{L(x)}$ where $L(x) = L_0 + L_1\|\nabla f(x)\|$ for some $\nu$,

$$f_* \leq f(y) \leq f(x) - \frac{\nu\|\nabla f(x)\|^2}{L(x)} + \frac{1}{2}\exp\left(\frac{L_1\nu\|\nabla f(x)\|}{L(x)}\right)\frac{\|\nabla f(x)\|^2\nu^2}{L(x)}.$$

Using the fact that $\frac{L_1\|\nabla f(x)\|}{L(x)} = \frac{L_1\|\nabla f(x)\|}{L_0 + L_1\|\nabla f(x)\|} \leq 1$, then by (19),

$$f_* \leq f(x) - \frac{\nu\|\nabla f(x)\|^2}{L(x)} + \frac{\|\nabla f(x)\|^2\nu^2\exp(\nu)}{2L(x)}$$
$$= f(x) - \frac{\nu\|\nabla f(x)\|^2}{L(x)}\left(1 - \frac{\nu\exp(\nu)}{2}\right)$$

Now setting $\nu$ such that $\nu\exp(\nu) = 1 \implies \nu \in (1/2, 1)$, we see that

$$f_* \leq f(x) - \frac{\nu\|\nabla f(x)\|^2}{2L(x)}.$$

Since $\nu > \frac{1}{2}$, then for all $x$,

$$\|\nabla f(x)\|^2 \leq \frac{2}{\nu}(L_0 + L_1\|\nabla f(x)\|)(f(x) - f_*) \leq 4(L_0 + L_1\|\nabla f(x)\|)(f(x) - f_*).$$

Now suppose $f(x) - f_* \leq \delta$, then the above inequality implies

$$\|\nabla f(x)\|^2 \leq 4(L_0 + L_1\|\nabla f(x)\|)\delta,$$

which further implies

$$\|\nabla f(x)\|^2 - 4L_1\delta\|\nabla f(x)\| - 4L_0\delta \leq 0.$$

Now using the quadratic formula, we can bound the gradient norm such that

$$\|\nabla f(x)\| \leq \frac{4L_1\delta + \sqrt{16L_1^2\delta^2 + 16L_0\delta}}{2}.$$

Hence, for all $x$ s.t. $f(x) - f^* \leq \delta$, by $\sqrt{a+b} \leq \sqrt{a} + \sqrt{b}$,

$$\|\nabla f(x)\| \leq 4L_1\delta + 2\sqrt{L_0\delta}.$$

Using the above inequality with the definition of $(L_0, L_1)$ non-uniform smoothness, we obtain that for all $y$ and $x$ s.t. $f(x) - f^* \leq \delta$,

$$\|\nabla f(x) - \nabla f(y)\| \leq \left(L_0 + L_1(4L_1\delta + 2\sqrt{L_0\delta})\right)\|y - x\|,$$

which implies that $f$ is locally $(L_0 + L_1(4L_1\delta + 2\sqrt{L_0\delta}))$-smooth.

$\square$

## B  Logistic Regression Glocally Smooth Results

*Proof of Proposition 3.* The Hessian matrix of the logistic regression objective is given by,

$$\nabla^2 \ell(w) = X^T D X,$$

where $D$ is an $n \times n$ diagonal matrix with diagonal elements $D_{i,i} = p_i (1 - p_i)$ where $p_i = \frac{1}{1 + \exp(-y_i \langle x_i, w \rangle)}$. Since $p_i \in [0, 1]$ have that $D_{i,i} \leq \frac{1}{4}$ and hence $\ell(w)$ is globally $\frac{\lambda_{\max}[X^T X]}{4}$-smooth. Let us upper-bound $D_{i,i}$ in a tighter way such that

$$D_{i,i} = p_i (1 - p_i) = \frac{1}{1 + \exp(-y_i \langle x_i, w \rangle)} \frac{\exp(-y_i \langle x_i, w \rangle)}{1 + \exp(-y_i \langle x_i, w \rangle)}.$$

Now since $\frac{1}{1+e^x} \leq 1$ for all $x$,

$$D_{i,i} \leq \frac{\exp(-y_i \langle x_i, w \rangle)}{1 + \exp(-y_i \langle x_i, w \rangle)}.$$

Using the fact that $\frac{x}{1+x} \leq \ln(1 + x)$ for all $x$,

$$D_{i,i} \leq \ln(1 + \exp(-y_i \langle x_i, w \rangle)),$$

and therefore

$$D_{i,i} \leq \sum_{i=1}^{n} \ln(1 + \exp(-y_i \langle x_i, w \rangle)) = \ell(w).$$

Defining $U := \ell(w) I_n$, we know that $D_{i,i} \leq U_{i,i}$ for all $i$. Using the above relation,

$$\nabla^2 \ell(w) = X^T D X \preceq U X^T X \preceq \ell(w) \lambda_{\max}[X^T X] I_n$$
$$\implies \|\nabla^2 \ell(w)\| \leq \ell(w) \lambda_{\max}[X^T X]$$

This implies that for *all* $w$, if $\ell(w) - \ell_* \leq \delta$, then $L_* \leq [\ell_* + \delta] \lambda_{\max}[X^T X]$.

$\square$

Note that by bounding $D_{i,i}$ by $\max_i \ln(1 + \exp(-y_i \langle x_i, w \rangle))$, we can get a tighter result than the one given above. However, we choose to use the looser bound in our proof as it leads to a more elegant result.

### B.1  Complexity of Logistic Regression Under Glocal Smoothness

To show Proposition 9, we first derive the $\delta$ minimizing the iteration complexity bound in Theorem 6 using the glocal logistic regression bound.

**Lemma 24.** Consider the function

$$h(\delta) = \frac{1}{4} \log\left(\frac{\Delta_0}{\delta}\right) + (\ell_* + \delta) \log\left(\frac{\delta}{\epsilon}\right), \tag{20}$$

with $\Delta_0 \geq \delta \geq \epsilon > 0$ and $\ell_* \geq 0$. Define $\xi = 1/4 - \ell_*$.

- Case 1: if $\xi \leq \epsilon$ the minimizer of this function is $\delta_* = \epsilon$.

- Case 2: if $\xi \geq \Delta_0 \log(\Delta_0 e/\epsilon)$ the minimizer of this function is $\delta_* = \Delta_0$.

- Case 3: otherwise the minimizer of this function is

$$\delta_* = \frac{\xi}{\omega}, \tag{21}$$

or equivalently

$$\delta_* = \frac{\epsilon}{e} \exp(\omega), \tag{22}$$

where $\omega$ is the principal branch of the Lambert $W$ function evaluated at $\xi e/\epsilon$ (which is positive since in this case $\xi > \epsilon > 0$).

*Proof.* We first note that if $\xi \leq 0$ then $\ell_* > 1/4$ and the solution is given by $\delta_* = \epsilon$. Thus in deriving the other situations below we will assume that $\xi > 0$. Taking the derivative of $h$ with respect to $\delta$ gives

$$h'(\delta) = -\frac{1}{4\delta} + \frac{\ell_*}{\delta} + \log \delta + 1 - \log \epsilon$$
$$= \log\left(\frac{\delta e}{\epsilon}\right) - \frac{\xi}{\delta},$$

and we note that both terms are strictly increasing in $\delta$ (for $\xi > 0$). Thus, to find a minimizer it is sufficient to check the directional derivatives at the boundaries and for stationary points in the interior.

- Case 1: at the left boundary of $\delta = \epsilon$ we have

$$h'(\epsilon) = 1 - \frac{\xi}{\epsilon}.$$

We have that $\epsilon$ is a minimizer if $h'(\epsilon) \geq 0$, which happens if $\xi \leq \epsilon$.

- Case 2: at the right boundary of $\delta = \Delta_0$ we have

$$h'(\Delta_0) = \log\left(\frac{\Delta_0 e}{\epsilon}\right) - \frac{\xi}{\Delta_0},$$

and we have $h'(\Delta_0) \leq 0$ if $\xi \geq \Delta_0 \log(\Delta_0 e/\epsilon)$

- Case 3: in this case we have a stationary point in the interior. Equating $h'(\delta)$ with 0 is equivalent to

$$\frac{\xi}{\delta} = \log\left(\frac{\delta e}{\epsilon}\right).$$

Applying the exponent function to both sides, we have

$$\exp\left(\frac{\xi}{\delta}\right) = \frac{\delta e}{\epsilon}.$$

Multiplying both sides by $\xi/\delta$ gives,

$$\frac{\xi}{\delta} \exp\left(\frac{\xi}{\delta}\right) = \frac{\xi e}{\epsilon}.$$

Now in order to find the minimizer of the above equation, we apply the Lambert $W$ function,

$$W\left(\frac{\xi}{\delta} \exp\left(\frac{\xi}{\delta}\right)\right) = W\left(\frac{\xi e}{\epsilon}\right),$$

and applying the defining property $(W(y) \exp(W(y)) = y$ for any $y)$ of the Lambert $W$ function on the left gives

$$\frac{\xi}{\delta} = W\left(\frac{\xi e}{\epsilon}\right),$$

where we note that $W(\xi e/\epsilon)$ is real and unique since $\xi e/\epsilon > 0$. Solving for $\delta$ gives

$$\delta = \frac{\xi}{W(\frac{\xi e}{\epsilon})},$$

or by re-arranging the Lambert equation $W(y)\exp(W(y)) = y$ as $W(y) = y/\exp(W(y))$ we equivalently have

$$\delta = \frac{\epsilon}{e}\exp\left(W\left(\frac{\xi e}{\epsilon}\right)\right).$$

$\square$

*Proof of Proposition 9.* Since $\ell$ is glocally $(L, L_*, \delta)$-smooth (by Lemma 3) and $\mu$-strongly convex, by Theorem 6 the iteration complexity of GD(LO) as a function of $\delta$ is

$$T(\delta) = \frac{L}{\mu}\log\left(\frac{\Delta_0}{\delta}\right) + \frac{L_*}{\mu}\log\left(\frac{\delta}{\epsilon}\right).$$

Now if use the bounds $L = \frac{1}{4}\|X\|^2$ and $L_* = (\ell_* + \delta)\|X\|^2$,

$$T(\delta) = \frac{\|X\|^2}{\mu}\left(\frac{1}{4}\log\left(\frac{\Delta_0}{\delta}\right) + (\ell_* + \delta)\log\left(\frac{\delta}{\epsilon}\right)\right) \tag{23}$$

$$= \frac{\|X\|^2}{\mu}h(\delta), \tag{24}$$

where $h(\delta)$ is given by (20). We obtain the different cases by plugging into $h(\delta)$ the minimizing values $\delta_*$ from Lemma 24. Case 1 and Case 2 are obtained by using $\delta = \epsilon$ and $\delta = \Delta_0$ (respectively) and using that $\log(1) = 0$. Case 3 is obtained using the two forms of $\delta_*$ from Case 3 of Lemma 24,

$$h(\delta_*) = \frac{1}{4}\log\left(\frac{\Delta_0}{\delta_*}\right) + (\ell_* + \delta_*)\log\left(\frac{\delta_*}{\epsilon}\right)$$

$$= \frac{1}{4}\log\left(\frac{\Delta_0 e}{\exp(\omega)\epsilon}\right) + \left(\ell_* + \frac{\xi}{\omega}\right)\log\left(\frac{\exp(\omega)}{e}\right) \qquad (\delta_* = \xi/\omega = \exp(\omega)\epsilon/e)$$

$$= \frac{1}{4}\log\left(\frac{\Delta_0}{\epsilon}\right) + \underbrace{\left(\frac{1}{4} - \ell_*\right)}_{\xi} - \omega\underbrace{\left(\frac{1}{4} - \ell_*\right)}_{\xi} + \xi - \frac{\xi}{\omega}$$

$$= \frac{1}{4}\log\left(\frac{\Delta_0}{\epsilon}\right) - \xi\omega + 2\xi - \frac{\xi}{\omega}$$

$$= \frac{1}{4}\log\left(\frac{\Delta_0}{\epsilon}\right) - \frac{\xi}{\omega}(\omega^2 - 2\omega + 1)$$

$$= \frac{1}{4}\log\left(\frac{\Delta_0}{\epsilon}\right) - \frac{\xi}{\omega}(\omega - 1)^2$$

$\square$

## C   Progress Guarantee Lemma

*Proof of Proposition 7.* Our goal is to show that steps along a descent direction do not leave the $\delta + f_*$-sublevel set as long as the step-size is not too large. Without loss of generality, assume that $d$ is a unit vector (otherwise replace $d$ with $\bar{d} = d/\|d\|_2$ and the proof is the same). Since $d$ is a descent direction, the directional derivative is strictly negative ($f$ is decreasing in this direction) at $\alpha = 0$. Thus, if some non-zero step-size

$$\alpha_* \leq \frac{1}{L_*}|\langle\nabla f(w_t), d\rangle|,$$

were to increase $f$, the sign of the directional derivative would have to change along the interval between 0 and $\alpha_*$. Suppose $f(w_t + \alpha_* d) \geq f(w_t)$ so that this is the case.

Because the directional derivative is continuous, it must have at least one zero between 0 and $\alpha_*$ by the intermediate value theorem. Let $\hat{\alpha}$ be the smallest $\alpha$ in $(0, \alpha_*)$ for which the directional derivative is zero. That is, it satisfies

$$\langle \nabla f(w_t + \hat{\alpha} d), d \rangle = 0.$$

Continuity of the gradient (and therefore the directional derivative) guarantees that $\hat{\alpha}$ exists and is strictly positive.

Our choice of $\hat{\alpha}$ also guarantees that the directional derivative is strictly negative for all smaller step-sizes, i.e. $\langle \nabla f(w_t + \alpha d), d \rangle < 0$ for all $\alpha \leq \hat{\alpha}$. Integrating implies,

$$f(w_t + \hat{\alpha} d) - f(w_t) = \int_0^{\hat{\alpha}} \langle \nabla f(w_t + rd), d \rangle \, dr < 0,$$

from which we deduce that $f(w_t + \hat{\alpha} d) < f(w_t)$. So, $w_t + \hat{\alpha} d$ is in the $\delta$ sub-level set. Now we need only apply $L_*$-smoothness along the chord between 0 and $\hat{\alpha}$ as follows. Let $g(r) = f(w_t + rd)$ and observe,

$$
\begin{aligned}
\alpha_* &\leq \frac{1}{L_*} |\langle \nabla f(w_t), d \rangle| \\
&= \frac{1}{L_*} |g'(0)| && \text{(By definition of the directional derivative.)} \\
&= \frac{1}{L_*} |g'(0) - g'(\hat{\alpha})| && \text{(Since $\hat{\alpha}$ is a stationary point of $g$.)} \\
&\leq |0 - \hat{\alpha}| \\
&= \hat{\alpha},
\end{aligned}
$$

where we have used Lemma 25 to extend $L_*$-smoothness of $f$ to $g$. But since $\alpha_* > \hat{\alpha}$, we obtain a contradiction. This completes the proof. $\qquad \square$

**Lemma 25.** Assume $f$ is $L_*$-smooth over some set $C$, and that $\|d\| = 1$. Let $g(\eta) = f(w + \eta d)$. Then $g(\eta)$ is $L_*$-smooth over $\{\eta : w + \eta d \in C\}$.

*Proof.*

$$|g'(\alpha) - g'(\beta)| = |\langle \nabla f(w + \alpha d), d \rangle - \langle \nabla f(w + \beta d), d \rangle|$$

Using the Cauchy–Schwarz inequality and our assumption that $\|d\| = 1$,

$$|g'(\alpha) - g'(\beta)| \leq \|\nabla f(w + \alpha d) - \nabla f(w + \beta d)\|.$$

By the $L_*$-smoothness of $f$ and our assumption that $\|d\| = 1$,

$$
\begin{aligned}
|g'(\alpha) - g'(\beta)| &\leq L_* \|(\beta - \alpha) d\| \\
&= L_* |\beta - \alpha|.
\end{aligned}
$$

$$\qquad \square$$

*Proof of Corollary 8.* For gradient descent at a point $w_t$, the next step is taken in the direction $d = -\nabla f(w_t)$. With step size $\frac{1}{L_*}$, $\eta_* \|d\|^2 = \frac{1}{L_*} \|\nabla f(w_t)/L_*\|^2 = \frac{1}{L_*} |\langle \nabla f(w_t), d \rangle|$. Hence, the condition $\eta_* \|d\|^2 \leq \frac{1}{L_*} |\langle \nabla f(w_t), d \rangle|$ in Proposition 7 is satisfied, so $f(w_{t+1}) - f_* \leq \delta$. Note that using a step size of $\frac{1}{L_*}$ guarantees progress of $f(w_{t+1}) \leq f(w_t) - \frac{1}{2L_*} \|\nabla f(w_t)\|^2$. Then since an exact line search will result in a lower value of $f$ than $f(w_{t+1})$ using $\frac{1}{L_*}$, this proves the result. $\qquad \square$

# D  Armijo Line Search Results

In this section, we give the proof of Theorem 10. We start by presenting more details about the line search procedure proposed in Section 4. If the initial step size $\tilde{\eta}_t$ on iteration $t$ does not satisfy 8, it is decreased by a factor $\beta$ where $0 < \beta < 1$ until the condition is satisfied. In particular, the step size $\eta_t$ selected by the Armijo condition is set as $\eta_t = \beta^{i_t} \tilde{\eta}_t$, where $i_t$ is the number of times the step size is decreased on iteration $t$. In this section, we focus on a variant of the backtracking line search that also makes use of forwardtracking (Fridovich-Keil & Recht, 2019). This modifies the line search to increase the step size on each iteration by a factor of $1/\beta$ until the condition (8) fails before performing the backtracking, allowing for selection of larger step sizes.

We now prove the following lemma, which is necessary for our proof of Theorem 10. It modifies the standard backtracking argument (Boyd & Vandenberghe, 2004) to account for the forward tracking and generalizes the result of Fridovich-Keil & Recht (2019) to allow $\alpha < 1/2$.

**Lemma 26.** Assume that $f$ is $\bar{L}$-smooth and $\mu$-strongly convex. For all $t$, for the iterates of gradient descent as defined in (1) with step-size $\eta_t$ selected according to (8) with forwardtracking, the guaranteed progress is

$$f(w_t) \leq f(w_{t-1}) - \frac{\beta\alpha}{\bar{L}}\|\nabla f(w_{t-1})\|^2, \tag{25}$$

which under strong convexity implies

$$f(w_t) - f_* \leq (1 - \frac{2\beta\alpha\mu}{\bar{L}})(f(w_{t-1}) - f_*)$$

where $0 < \alpha < 1/2$ and $0 < \beta < 1$. This implies that $f(w_t)$ decreases monotonically for all $t$. It also implies that if we start from $\bar{w}_0$ we require $t = \frac{\bar{L}}{2\beta\alpha\mu} \log\left(\frac{f(\bar{w}_0) - f_*}{\bar{\epsilon}}\right)$ iterations to have $f(w_t) - f_* \leq \bar{\epsilon}$.

*Proof.* By the $\bar{L}$-smoothness of $f$ and the gradient descent update,

$$f(w_t) \leq f(w_{t-1}) + \langle \nabla f(w_{t-1}), w_t - w_{t-1}\rangle + \frac{\bar{L}}{2}\|w_t - w_{t-1}\|^2$$

$$\leq f(w_{t-1}) - \eta_t\|\nabla f(w_{t-1})\|^2 + \frac{\bar{L}\eta_t^2}{2}\|\nabla f(w_{t-1})\|^2.$$

$$\leq f(w_{t-1}) - \eta_t(1 - \frac{\bar{L}\eta_t}{2})\|\nabla f(w_{t-1})\|^2.$$

Combining the above with the Armijo condition 8 with forwardtracking, we get that the step size $\eta_t$ returned by the line search satisfies $\eta_t \geq \frac{\beta}{\bar{L}}$. Now since the step size must satisfy 8, this implies that

$$f(w_t) \leq f(w_{t-1}) - \frac{\beta\alpha}{\bar{L}}\|\nabla f(w_{t-1})\|^2.$$

Now using the PL inequality, which is implied by strong convexity,

$$f(w_t) \leq f(w_{t-1}) - \frac{2\beta\alpha\mu}{\bar{L}}(f(w_{t-1}) - f_*).$$

Subtracting $f_*$ from both sides gives the result. $\qquad\square$

*Proof of Theorem 10.* We follow the analysis structure outlined in Section 3.5:

1. Our region of interest under glocal smoothness is the set $\{w \mid f(w) - f_* \leq \delta\}$.

2. Let $t_\delta$ be the first iteration such that $f(w_{t_\delta}) - f_* \leq \delta$. From the monotonic decrease of $f(w_t)$, it follows that $w_t$ stays in the region of interest for subsequent $t$.

3. Initialized at $w_0$, using Lemma 26 with $\bar{L} = L$, GD with Armijo line search is guaranteed to satisfy $f(w_t) - f_* \leq \delta$ after $t_\delta = \left\lceil \frac{L}{2\beta\alpha\mu} \log\left(\frac{f(w_0)-f_*}{\delta}\right) \right\rceil$ iterations.

4. We start by noting that Proposition 7 implies that for all $t \geq t_\delta$ and $\eta \leq 1/L_*$, the gradient descent update

$$w_t = w_{t-1} - \eta\nabla f(w_{t-1})$$

satisfies $f(w_t) - f_* \leq \delta$. Thus, $L_*$ smoothness holds between $w_t$ and $w_{t-1}$, implying that for $\eta \leq 1/L_*$,

$$f(w_t) \leq f(w_{t-1}) - \eta(1 - \frac{L_*\eta}{2})\|\nabla f(w_{t-1})\|^2.$$

As a result, the Armijo condition eq. (8) is satisfied for all $\eta \leq 1/L_*$. The Armijo line search with backtracking parameter $\beta$ is thus guaranteed to return $\eta_t \geq \frac{\beta}{L_*}$ for all $t \geq t_\delta$. Therefore 25 holds with $\bar{L} = L_*$ and we can use Lemma 26 with $\bar{L} = L_*$ and $\bar{\epsilon} = \epsilon$ to obtain that GD with Armijo line search starting from $\bar{w}_0 = w_{t_\delta}$ is guaranteed to have $f(w_t) - f_* \leq \epsilon$ after another $t = \left\lceil \frac{L_*}{2\beta\alpha\mu} \log\left(\frac{f(w_{t_\delta})-f_*}{\epsilon}\right) \right\rceil$ iterations.

5. Using that $f(w_{t_\delta}) - f_* \leq \delta$, we can conclude that it is sufficient to have $t = \left\lceil \frac{L_*}{2\beta\alpha\mu} \log\left(\frac{\delta}{\epsilon}\right) \right\rceil$ additional iterations.

$\square$

# E   Polyak Step Size Convergence Results

In this section we give the proof of Theorem 11. We also give an example where the Polyak step size leads to an increase in the function value, and also analyze the Polyak step size under glocal smoothness of the iterates (Theorem 28). Our analyses exploit the following result (Hazan & Kakade, 2019, Lemma 1, Lemma 2) for gradient descent with the classic Polyak step size (Polyak, 1987).

**Lemma 27.** Assume that $f$ is $\bar{L}$-smooth and $\mu$-strongly convex. For all $t$ the iterates of gradient descent as defined in (1) with step-size $\eta_t$ given by (9) satisfy

$$\|w_t - w_*\|^2 \leq (1 - \frac{\mu}{4\bar{L}})\|w_{t-1} - w_*\|^2$$

This implies that $\|w_t - w_*\|^2$ decreases monotonically for all $t$. It also implies that if we start from $\bar{w}_0$ we require $t = \frac{4\bar{L}}{\mu} \log\left(\frac{\|\bar{w}_0 - w_*\|^2}{\bar{\epsilon}}\right)$ iterations to have $\|w_t - w_*\|^2 \leq \bar{\epsilon}$.

## E.1   Polyak Convergence Proof with Glocal Assumption in Function Values

*Proof of Theorem 11.* We follow the analysis structure outlined in Section 3.5:

1. Our region of interest under glocal smoothness is $\{w | f(w) - f_* \leq \delta\}$.

2. The Polyak step does not guarantee decrease in the function value, but we remain in the region of interest once $\|w_t - w_*\|$ becomes sufficiently small. Let $t_\delta$ be the first iteration where $\frac{L}{2}\|w_{t_\delta} - w_*\|^2 \leq \delta$. By global smoothness and the fact that $\nabla f(w_*) = 0$,

$$f(w) - f_* \leq \frac{L}{2}\|w - w_*\|^2,$$

and thus, $f(w_{t_\delta}) - f_* \leq \delta$. Further by the monotonicity of $\|w_t - w_*\|$, we have $f(w_t) - f_* \leq \delta$ for all $t \geq t_\delta$.

3. Initialized at $w_0$, using corollary 27 with $\bar{L} = L$, GD(Polyak) is guaranteed to have $\frac{L}{2}\|w_t - w_*\|^2 \leq \delta$ after $t_\delta = \lceil \frac{4L}{\mu} \log\left(\frac{L}{2}\frac{\|w_0 - w_*\|^2}{\delta}\right)\rceil$ iterations.

4. By $L_*$ smoothness within the region of interest we have $f(w_t) - f_* \leq \epsilon$ if we have $\frac{L_*}{2}\|w_t - w_*\|^2 \leq \epsilon$ for $t \geq t_\delta$. Initialized at $w_{t_\delta}$, using corollary 27 with $\bar{L} = L_*$, we have that $\frac{L_*}{2}\|w_t - w_*\|^2 \leq \epsilon$ after an additional $t = \lceil \frac{4L_*}{\mu} \log\left(\frac{L_*}{2}\frac{\|w_{t_\delta} - w_*\|^2}{\epsilon}\right)\rceil$ iterations.

5. Since $\frac{L}{2}\|w_{t_\delta} - w_*\|^2 \leq \delta$, it is sufficient to have $t = \lceil \frac{4L_*}{\mu} \log\left(\frac{L_*}{L}\frac{\delta}{\epsilon}\right)\rceil$ additional iterations.

$\square$

## E.2 Example of Polyak Step Size Increasing Function Value

In this section, we give an example showing that gradient descent with the Polyak step size can increase the function value while decreasing the 2-norm distance to the minimizer. Consider the function

$$f(w_0, w_1) = \frac{1}{2}w_0^2 + \frac{1}{40}w_1^2.$$

Note that the minimizer is $f_* = 0$, and the minimum is $w_* = (0,0)$. The gradient of $f$ is given by

$$g(w_0, w_1) = w_0 + \frac{1}{20}w_1.$$

Consider the iterate $\hat{w} = (0.05, 1)$. Then $f(\hat{w}) \approx 0.02625$ and $\|\hat{w} - w_*\|^2 \approx 1.00125$. The Polyak step size at this step is $\hat{\eta} \approx 5.25$, and the next iterate of gradient descent is $w' = (-0.2125, 0.7375)$. With this step, $f(w') \approx 0.03617$ and $\|w' - w_*\|^2 \approx 0.7675$. Thus, gradient descent with the Polyak step size can increase the function value while decreasing the 2-norm distance to the minimizer.

## E.3 Polyak Convergence Proof with Glocal Assumption in Iterates

**Theorem 28.** Assume that $f$ is glocally $(L, L_*, \delta)$-smooth in iterates and $\mu$-strongly convex. For all $t \geq 0$, let $w_t$ be the iterates of gradient descent as defined in (1) with step-size $\eta_t$ given by (9). Then $\|w_T - w_*\|^2 \leq \epsilon$ for all

$$T \geq 4\left(\frac{L}{\mu}\log\left(\frac{\|w_0 - w_*\|^2}{\delta}\right) + \frac{L_*}{\mu}\log\left(\frac{\delta}{\epsilon}\right)\right).$$

*Proof.* We follow the analysis structure outlined in Section 3.5:

1. Our region of interest under glocal smoothness of the iterates is $\{w | \|w - w_*\|^2 \leq \delta\}$.

2. Let $t_\delta$ be the first iteration such that $\|w_{t_\delta} - w_*\|^2 \leq \delta$. It follows from the monotonicity of $\|w_t - w_*\|$, that all iterates $w_t$ for $t \geq t_\delta$ stay in the region of interest.

3. Initialized at $w_0$, using Lemma 27 with $\bar{L} = L$, GD(Polyak) is guaranteed to have $\|w_t - w_*\|^2 \leq \delta$ after $t_\delta = \lceil \frac{4L}{\mu}\log\left(\frac{\|w_0 - w_*\|^2}{\delta}\right)\rceil$ iterations.

4. Initialized at $w_{t_\delta}$, using Lemma 27 with $\bar{L} = L_*$, we have that $\|w_t - w_*\|^2 \leq \epsilon$ after an additional $t = \lceil \frac{4L}{\mu}\log\left(\frac{\|w_{t_\delta} - w_*\|^2}{\epsilon}\right)\rceil$ iterations.

5. Since $\|w_{t_\delta} - w_*\|^2 \leq \delta$, it is sufficient to have $t = \lceil \frac{4L_*}{\mu}\log\left(\frac{\delta}{\epsilon}\right)\rceil$ additional iterations.

$\square$

## F    AdGD Convergence Results

In this section we give the proof of Theorem 13, which makes use of an existing convergence result for AdGD (Malitsky & Mishchenko, 2020, Theorem 2).

**Lemma 29.** Assume that $f$ is $\bar{L}$-smooth and $\mu$-strongly convex. For all $t > 2$ the iterates of gradient descent as defined in (1) with step-size $\eta_t$ given by (10) satisfy

$$\Phi_{t+1} \leq \left(1 - \frac{\mu}{4\bar{L}}\right)\Phi_t$$

where $\Phi_t = \|w_t - w_*\|^2 + \frac{1}{2}(1 + \frac{2\mu}{\bar{L}})\|w_t - w_{t-1}\|^2 + 2\eta_{t-1}(1 + \theta_{t-1})(f(w_{t-1}) - f_*))$.

This implies that $\Phi_t$ decreases monotonically for all $t > 2$. It also implies that we require $t = O\left(\frac{\bar{L}}{\mu}\log\left(\frac{1}{\bar{\epsilon}}\right)\right)$ iterations to have $\Phi_t \leq \bar{\epsilon}$. Note that the proof of the above result uses the bound in Lemma 41. Lemma 42 shows that this bound also holds in the local region with smoothness $L_*$, which is required for step 4.

*Proof of Theorem 13.* We follow the analysis structure outlined in Section 3.5:

1. Our region of interest under glocal smoothness of the iterates is $\{w \mid \|w - w_*\|^2 \leq \delta\}$.

2. Let $t^{'}$ be the first iteration such that $\Phi_{t'} \leq \delta$ and $t^{'} > 2$. Now let $t_\delta = t^{'} + 2$. Since $\|w_t - w_*\|^2 \leq \Phi_t$ for all $t$, it follows from the monotonicity of $\Phi_t$ that all $t \geq t_\delta - 1$ stay in the region of interest.

3. Initialized at $w_0$, using Lemma 29 with $\bar{L} = L$, AdGD is guaranteed to have $\|w_{t_\delta} - w_*\|^2 \leq \Phi_{t_\delta} \leq \delta$ after $t_\delta = O\left(\frac{L}{\mu}\log\left(\frac{\Phi_3}{\delta}\right)\right)$ iterations.

4. Initialized at $w_{t_\delta}$, using Lemma 29 with $\bar{L} = L_*$, we have that $\|w_t - w_*\|^2 \leq \Phi_t \leq \epsilon$ after another $t = O\left(\frac{L_*}{\mu}\log\left(\frac{\Phi_{t_\delta}}{\epsilon}\right)\right)$ iterations.

5. Since $\Phi_{t_\delta} \leq \delta$, it is sufficient to have $t = O\left(\frac{L_*}{\mu}\log\left(\frac{\delta}{\epsilon}\right)\right)$ additional iterations.

$\square$

## G    Convex Objectives

In order to analyze GD(LO) in convex settings, we require an iteration complexity of GD(LO) for smooth and convex functions. We give such a result next, following an argument similar to previous work (Beck & Tetruashvili, 2013, Section 3).

**Lemma 30.** Assume that $f$ is $\bar{L}$-smooth. For all $t$, for the iterates of gradient descent as defined in (1) with step-size $\eta_t$ given line optimization (2), the guaranteed progress is

$$f(w_t) \leq f(w_{t-1}) - \frac{1}{2\bar{L}}\|\nabla f(w_{t-1})\|^2,$$

Furthermore, if $f$ is convex and coercive, then this implies that

$$f(w_t) - f_* \leq \frac{2\bar{L}R^2(\rho)}{t + 4}$$

where

$$R^2(\rho) = \max_{w \in \mathbb{R}^d, w_* \in W_*}\{\|w - w_*\|^2 : f(w) \leq \rho)\}.$$

*Proof.* Using an exact line search with $\bar{L}$-smoothness guarantees descent such that,

$$f(w_t) \leq f(w_{t-1}) - \frac{1}{2\bar{L}}\|\nabla f(w_{t-1})\|^2. \tag{26}$$

Now by convexity and the Cauchy-Schwartz inequality, for any $w_* \in W_*$:

$$\begin{aligned}
f(w_{t-1}) - f_* &\leq \langle \nabla f(w_{t-1}), w_{t-1} - w_* \rangle \\
&\leq \|\nabla f(w_{t-1})\|\|w_{t-1} - w_*\| \\
&\leq \|\nabla f(w_{t-1})\| \max_{w \in \mathbb{R}^d} \{\|w - w_*\| : f(w) \leq f(w_{t-1})\} \\
&\leq \|\nabla f(w_{t-1})\| \max_{w \in \mathbb{R}^d} \{\|w - w_*\| : f(w) \leq f(w_0)\},
\end{aligned}$$

where we have used that monotonicity implies that $f(w_{t-1}) \leq f(w_0)$. Now using that $f(w_0) \leq \rho$, we obtain

$$f(w_{t-1}) - f_* \leq \|\nabla f(w_{t-1})\|\sqrt{R^2(\rho)}.$$

We note that since $f$ is coercive, the level sets of $f$ are compact and $R^2(\rho) < \infty$. Combining the above with (26) we have

$$f(w_t) \leq f(w_{t-1}) - \frac{1}{2\bar{L}R^2(\rho)}(f(w_{t-1}) - f_*)^2.$$

Now, by $\bar{L}$-smoothness, we have that:

$$f(w_0) - f_* \leq \frac{\bar{L}}{2}\|w_0 - w_*\|^2 \leq \frac{\bar{L}}{2}R^2(\rho) = \left(\frac{1}{4}\right)2\bar{L}R^2(\rho)$$

Then by Lemma 43 with $m = 4$ and $\gamma = \frac{1}{2\bar{L}R^2(\rho)}$:

$$f(w_t) - f_* \leq \frac{2\bar{L}R^2(\rho)}{t+4}$$

as required. □

This result implies that $f(w_t) - f_*$ is less than a value $\bar{\epsilon} > 0$ beginning from $\bar{w}_0$ for all $t$ satisfying

$$t \geq O\left(\frac{\bar{L}}{\bar{\epsilon}}R^2(\rho)\right), \tag{27}$$

for any $\rho \geq f(\bar{w}_0)$.

### G.1 Globally Convex and Locally Strongly Convex

*Proof of Theorem 14.* We use the two-phase analysis strategy of Section 3.5. Comparing to the analysis of GD(LO) for glocally-smooth and globally strongly-convex functions, all the steps are identical except Step 3:

3. Initialized at $w_0$, using Lemma 30 with $\bar{L} = L$, GD(LO) is guaranteed to have $f(w_t) - f_* \leq \delta$ after $t_\delta = O\left(\frac{L}{\delta}R^2(f(w_0))\right)$ iterations.

□

### G.2 Globally Convex and Locally Convex

**Theorem 31.** Assume that $f$ is glocally $(L, L_*, \delta)$-smooth. Additionally, assume that $f$ is globally convex and coercive. For all $t \geq 0$, let $w_t$ be the iterates of gradient descent as defined in (1) with step-size $\eta_t$ given by (2). Then $f(w_T) - f_* \leq \epsilon$ for all

$$T \geq O\left(\frac{L}{\delta}R^2(f(w_0)) + \frac{L_*}{\epsilon}R^2(\delta + f_*)\right).$$

*Proof.* We only need to specify Steps 4 and 5 as the other steps are identical to the analysis of GD(LO) in the globally convex and locally strongly-convex case:

4. Initialized at $w_{t_\delta}$, using Corollary 8 Lemma 30 with $\bar{L} = L_*$, we have that $f(w_t) - f_* \leq \epsilon$ after another $t = O\left(\frac{L_*}{\epsilon}R^2(f(w_{t_\delta}))\right)$ iterations.

5. Using that $f(w_{t_\delta}) - f_* \leq \delta$, it is sufficient to have $t = O\left(\frac{L_*}{\epsilon}R^2(\delta + f_*)\right)$ additional iterations.

$\square$

## H  Coordinate Descent Results

In this section we give the proofs of Theorem 17 and Theorem 18. We first review a standard result regarding the convergence rate of random and greedy coordinate descent (Nesterov, 2012; Nutini et al., 2015).

**Lemma 32.** Assume that $f$ is coordinate-wise $\bar{L}$-smooth, then we have a coordinate-wise variant of the descent lemma

$$f(u + \alpha e_j) \leq f(u) + \alpha \nabla_j f(u) + \frac{\bar{L}}{2}|\alpha|^2.$$

It follows that for all $t$, for the iterates of coordinate descent as defined in (13) with step-size $\eta_t$ given by $1/\bar{L}$ or exact line optimization (14), we have guaranteed progress of

$$f(w_{t+1}) \leq f(w_t) - \frac{1}{2\bar{L}}(\nabla_{j_t} f(w_t))^2. \tag{28}$$

Furthermore, if we use uniform sampling and $f$ is $\mu$-strongly convex, then this implies that

$$E[f(w_t)] - f_* \leq \left(1 - \frac{\mu}{d\bar{L}}\right)(f(w_{t-1}) - f_*).$$

Alternately, if we use greedy selection and $f$ is $\mu_1$-strongly convex in the 1-norm, then this implies that

$$f(w_t) - f_* \leq \left(1 - \frac{\mu_1}{\bar{L}}\right)(f(w_{t-1}) - f_*).$$

Observe that (28) implies that $f(w_t) - f_*$ decreases monotonically for all $t$. Further, if we start from $\bar{w}_0$, we require $t = \frac{d\bar{L}}{\mu}\log\left(\frac{f(\bar{w}_0) - f_*}{\bar{\epsilon}}\right)$ iterations to have $E[f(w_t)] - f_* \leq \bar{\epsilon}$ with random sampling, whereas we require $t = \frac{\bar{L}}{\mu_1}\log\left(\frac{f(\bar{w}_0) - f_*}{\bar{\epsilon}}\right)$ iterations to have $f(w_t) - f_* \leq \bar{\epsilon}$ with greedy selection.

We next argue that, within the $\delta$ sublevel set, that coordinate descent methods with exact LO achieve (28) with $\bar{L} = L_*$.

**Corollary 33.** Assume that $w_t$ satisfies $f(w_t) - f_* \leq \delta$. Additionally, assume that $f$ is coordinate-wise $(L, L_*, \delta)$ glocally smooth. Then for coordinate descent with steps defined in (13) with step size of $\frac{1}{L_*}$, $f(w_{t+1}) - f_* \leq \delta$.

Furthermore, if (14) is used to select the step size using some $j_t$, the guaranteed progress is at least $f(w_{t+1}) \leq f(w_t) - \frac{1}{2L_*}(\nabla_{j_t} f(w_t))^2$.

*Proof.* For coordinate descent at a point $w_t$, the next step is taken in the direction $d = -\nabla_{j_t} f(w_t) e_{j_t}$. With step size $\frac{1}{L_*}$ and recalling that $e_{j_t}$ refers to a vector with entry 1 at index $j_t$ and all other entries 0, it is straightforward to verify that the condition $\eta_* \|d\|^2 \leq \frac{1}{L_*} |\langle \nabla f(w_t), d \rangle|$ in Proposition 7 is satisfied, so $f(w_{t+1}) - f_* \leq \delta$. Note that using a step size of $\frac{1}{L_*}$ guarantees progress of $f(w_{t+1}) \leq f(w_t) - \frac{1}{2L_*}(\nabla_{j_t} f(w_t))^2$ for some $j_t$. Then since an exact line search will make at least as much progress as using $\frac{1}{L_*}$, this proves the result. $\qquad\square$

*Proof of Theorem 17.* We follow the analysis structure outlined in Section 3.5, modifying Steps 2-5 to account for the stochasticity in the update.

1. Our region of interest under coordinate-wise glocal smoothness is $\{w | f(w) - f_* \leq \delta\}$.

2. Let $\rho = 1 - \frac{\mu}{dL}$ where we note that $0 < \rho < 1$. Following a similar argument to prior work (Konečný & Richtárik, 2017, Theorem 5), using Markov's inequality we have for any $\lambda > 0$,

$$P(f(w_t) - f_* > \lambda(f(w_0) - f_*)) \leq \frac{E[f(w_t)] - f_*}{\lambda(f(w_0) - f_*)}$$

$$\leq \frac{\rho^t}{\lambda}. \quad \text{(Using the convergence guarantee)}$$

Choosing $\lambda = \frac{\delta}{f(w_0) - f_*}$ we obtain

$$P(f(w_t) - f_* > \delta) \leq \frac{\rho^t [f(w_0) - f_*]}{\delta}.$$

For a probability $\zeta$ such that $0 < \zeta < 1$, the right side above is bounded above by $\zeta$ for

$$t = t_\delta := \left\lceil \frac{dL}{\mu} \log \left( \frac{f(w_0) - f_*}{\delta \zeta} \right) \right\rceil.$$

Thus, we have

$$P(f(w_{t_\delta}) - f_* \leq \delta) \geq 1 - \zeta.$$

Then with probability $1 - \zeta$, there is a first iteration $t_\delta$ such that $f(w_{t_\delta}) - f_* \leq \delta$. In the event where $f(w_{t_\delta}) - f_* \leq \delta$, it follows from monotonicity of $f(w_t)$ that all subsequent $t$ stay in the region of interest.

3. Initialized at $w_0$, using Lemma 32 with $\bar{L} = L$, random coordinate descent satisfies $f(w_t) - f_* \leq \delta$ after $t_\delta = \lceil \frac{dL}{\mu} \log \left( \frac{f(w_0) - f_*}{\delta \zeta} \right) \rceil$ iterations with probability $1 - \zeta$.

4. Define $X_\delta$ as the event that $f(w_{t_\delta}) - f_* \leq \delta$. Initialized at $w_{t_\delta}$, using Corollary 33 and Lemma 32 with $\bar{L} = L_*$, we have that $E[f(w_t) \mid X_\delta] - f_* \leq \epsilon$ after another $t = \lceil \frac{dL_*}{\mu} \log \left( \frac{f(w_{t_\delta}) - f_*}{\epsilon} \right) \rceil$ iterations.

5. Conditioned on $X_\delta$, $f(w_{t_\delta}) - f_* \leq \delta$. Hence, it is sufficient to have $t = \lceil \frac{dL_*}{\mu} \log \left( \frac{\delta}{\epsilon} \right) \rceil$ additional iterations.

$\qquad\square$

*Proof of Theorem 18.* We follow the analysis structure outlined in Section 3.5:

1. Our region of interest under coordinate-wise glocal smoothness is $\{w | f(w) - f_* \leq \delta\}$.

2. Let $t_\delta$ be the first iteration such that $f(w_{t_\delta}) - f_* \leq \delta$. It follows from monotonicity of $f(w_t)$ that all subsequent $t$ stay in the region of interest.

3. Initialized at $w_0$, using Lemma 32 with $\bar{L} = L$, coordinate descent with greedy coordinate selection is guaranteed to have $f(w_t) - f_* \leq \delta$ after $t_\delta = \lceil \frac{L}{\mu_1} \log \left( \frac{f(w_0) - f_*}{\delta} \right) \rceil$ iterations.

4. Initialized at $w_{t_\delta}$, using Corollary 33 and Lemma 32 with $\bar{L} = L_*$, we have that $f(w_t) - f_* \leq \epsilon$ after an additional $t = \lceil \frac{L_*}{\mu_1} \log \left( \frac{f(w_{t_\delta}) - f_*}{\epsilon} \right) \rceil$ iterations.

5. Using that $f(w_{t_\delta}) - f_* \leq \delta$, it is sufficient to have $t = \lceil \frac{L_*}{\mu_1} \log \left( \frac{\delta}{\epsilon} \right) \rceil$ additional iterations.

$\square$

# I Stochastic Gradient Descent Convergence Results

The following result is implied by existing work (Mishkin, 2020, Theorem 9) on using a stochastic line search under the interpolation assumption. Earlier work (Vaswani et al., 2019, Theorem 1) also gives a similar result.

**Lemma 34.** Assume that $f$ is $\mu$-strongly convex and that each $f_i$ is convex and $\bar{L}_{\max}$-smooth. Finally, assume that $f$ satisfies the interpolation assumption (19). Then stochastic gradient descent with the stochastic Armijo line search with $c = \frac{1}{2}$, $\beta = \frac{1}{2}$, and $\eta_{\max} \geq 1/\bar{L}$ satisfies

$$E[\|w_t - w_*\|^2] \leq \left( 1 - \frac{\mu}{2\bar{L}_{\max}} \right) \|w_{t-1} - w_*\|^2.$$

In addition, $\|w_t - w_*\|^2 \leq \|w_{t-1} - w_*\|^2$ for all $t$.

This result implies that if we start from $\bar{w}_0$ we require $t = \frac{2\bar{L}_{\max}}{\mu} \log \left( \frac{\|\bar{w}_0 - w_*\|^2}{\bar{\epsilon}} \right)$ iterations to have $E[\|w_t - w_*\|^2] \leq \bar{\epsilon}$. Further, note that the final comment implies that $\|w_t - w_*\|^2$ decreases monotonically, despite the algorithm being stochastic. Also, note that the proof of the above uses Lemma 41, and Lemma 42 verifies that Lemma 41 holds in the local region with smoothness $L_*$.

*Proof of Theorem 20.* We follow the analysis structure outlined in Section 3.5, modifying Steps 2-5 to reflect the stochastic nature of the method.

1. Our region of interest under glocal smoothness of the iterates is $\{w | \|w - w_*\|^2 \leq \delta\}$.

2. Let $\rho = 1 - \frac{\mu}{2\bar{L}_{\max}}$ where we note that $0 < \rho < 1$. Following a similar argument to prior work (Konečný & Richtárik, 2017, Theorem 5), using Markov's inequality we have for any $\lambda > 0$,

$$P(\|w_t - w_*\|^2 > \lambda \|w_0 - w_*\|^2) \leq \frac{E[\|w_t - w_*\|^2]}{\lambda \|w_0 - w_*\|^2} \leq \frac{\rho^t}{\lambda}.$$

Choosing $\lambda = \frac{\delta}{\|w_0 - w_*\|^2}$ we obtain

$$P(\|w_t - w_*\|^2 > \delta) \leq \frac{\rho^t \|w_0 - w_*\|^2}{\delta}.$$

For a probability $\zeta$ such that $0 < \zeta < 1$, the right side above is bounded above by $\zeta$ for

$$t_\delta = \left\lceil \frac{2L_{\max}}{\mu} \log \left( \frac{\|w_0 - w_*\|^2}{\delta \zeta} \right) \right\rceil.$$

Thus, we have

$$P(\|w_{t_\delta} - w_*\|^2 \leq \delta) \geq 1 - \zeta.$$

Then with probability $1 - \zeta$, there is a first iteration $t_\delta$ such that $\|w_{t_\delta} - w_*\|^2 \leq \delta$. It follows from Lemma 34 that all subsequent $t$ stay in the region of interest.

3. Initialized at $w_0$, using Lemma 34 with $\bar{L} = L_{\max}$, stochastic gradient descent satisfies $\|w_t - w_*\|^2 \leq \delta$ after $t_\delta = \left\lceil \frac{2L_{\max}}{\mu} \log\left(\frac{\|w_0 - w_*\|^2}{\delta\zeta}\right) \right\rceil$ iterations with probability $1 - \zeta$. Here we have used that $\eta_{\max} \geq \frac{1}{L_{\max,*}} \geq \frac{1}{L_{\max}}$.

4. Define $X_\delta$ as the event that $\|w_{t_\delta} - w_*\| \leq \delta$. Initialized at $w_{t_\delta}$, using Lemma 34 with $\bar{L} = L_{\max,*}$, and noting that we assume $\eta_{\max} \geq \frac{1}{L_{\max,*}}$, we have that $E[\|w_T - w_*\| \mid X_\delta] \leq \epsilon$ after another $t = \lceil \frac{2L_{\max,*}}{\mu} \log\left(\frac{\|w_{t_\delta} - w_*\|^2}{\epsilon}\right) \rceil$ iterations.

5. Conditioned on $X_\delta$, $\|w_{t_\delta} - w_*\|^2 \leq \delta$. Hence, it is sufficient to have $t = \lceil \frac{2L_{\max,*}}{\mu} \log\left(\frac{\delta}{\epsilon}\right) \rceil$ additional iterations.

$\square$

# J  Acceleration Convergence Results

**Lemma 35.** If $f$ is $\mu$-strongly convex, then the maximum step-size satisfying the Armijo condition in eq. (18) is bounded as,

$$\sup\left\{\eta > 0 : f(y_t - \eta\nabla f(y_t)) \leq f(y_t) - \frac{\eta}{2}\|\nabla f(y_t)\|^2\right\} \leq \frac{1}{\mu}.$$

Moreover, if $\eta = 1/\mu$, then $w_{t+1} = y_t - \eta_t\nabla f(y_t)$ satisfies $f(w_{t+1}) = f(w_*)$.

*Proof.* Since $f$ is strongly convex,

$$f(y_t - \eta\nabla f(y_t)) \geq f(y_t) - \eta\left(1 - \frac{\eta \cdot \mu}{2}\right)\|\nabla f(y_t)\|^2.$$

If $\eta > 1/\mu$, then $\left(1 - \frac{\eta\cdot\mu}{2}\right) < 1/2$ and we deduce,

$$f(y_t - \eta\nabla f(y_t)) > f(y_t) - \frac{\eta}{2}\|\nabla f(y_t)\|^2,$$

implying that the Armijo condition cannot hold.

Now, suppose that $\eta = 1/\mu$. Since the Armijo condition is satisfied, we have

$$f(w_{t+1}) \leq f(y_t) - \frac{1}{2\mu}\|\nabla f(y_t)\|_2^2.$$

However, strong convexity implies

$$f(w_*) \geq f(y_t) - \frac{1}{2\mu}\|\nabla f(y_t)\|_2^2.$$

We deduce that $f(w_{t+1}) = f(w_*)$ as claimed.  $\square$

**Lemma 36.** Let $w_{-1} = w_0$ and recall that $q_t = \eta_t \cdot \mu$. Then the acceleration scheme given in eq. (17) is equivalent to the following update,

$$
\begin{aligned}
y_t &= w_t + \left(\frac{1 - \sqrt{q_{t-1}}}{1 + \sqrt{q_t}}\right)\left(\frac{\eta_t}{\eta_{t-1}}\right)^{1/2}(w_t - w_{t-1}) \\
w_{t+1} &= y_t - \eta_t\nabla f(y_t)
\end{aligned}
\tag{29}
$$

*Proof.* We start by expressing the $z_t$ sequence in terms of $w_t$ as follows,

$$
\begin{aligned}
\sqrt{q_t} z_{t+1} &= \sqrt{q_t}(1 - \sqrt{q_t})z_t + q_t(y_t - \frac{1}{\mu}\nabla f(y_t)) \\
&= \sqrt{q_t}(1 - \sqrt{q_t})z_t + (q_t - 1)y_t + (y_t - \eta_t \nabla f(y_t)) \\
&= \sqrt{q_t}(1 - \sqrt{q_t})z_t + (q_t - 1)y_t + w_{t+1} \\
&= \sqrt{q_t}(1 - \sqrt{q_t})z_t + (q_t - 1)\left(w_t + \frac{\sqrt{q_t}}{1 + \sqrt{q_t}}(z_t - w_t)\right) + w_{t+1} \\
&= \left((q_t - 1) - \frac{\sqrt{q_t}(q_t - 1)}{1 + \sqrt{q_t}}\right)w_t + w_{t+1} \\
&= \left((\sqrt{q_t} - 1)(\sqrt{q_t} + 1) - \sqrt{q_t}(\sqrt{q_t} - 1)\right)w_t + w_{t+1} \\
&= w_{t+1} - (1 - \sqrt{q_t})w_t.
\end{aligned}
$$

As a result, we obtain,

$$
\sqrt{q_{t-1}} z_t = w_t - (1 - \sqrt{q_{t-1}})w_{t-1}.
$$

Plugging this expression into the definition of $y_t$ allows us to eliminate $z_t$,

$$
\begin{aligned}
y_t &= w_t + \frac{\sqrt{q_t}}{1 + \sqrt{q_t}}(z_t - w_t) \\
&= w_t + \frac{\sqrt{q_t}}{1 + \sqrt{q_t}}\frac{1}{\sqrt{q_{t-1}}}\left(\sqrt{q_{t-1}}z_t - \sqrt{q_{t-1}}w_t\right) \\
&= w_t + \frac{\sqrt{q_t}}{1 + \sqrt{q_t}}\frac{1 - \sqrt{q_{t-1}}}{\sqrt{q_{t-1}}}(w_t - w_{t-1}).
\end{aligned}
$$

Using $q_t/q_{t-1} = \eta_t/\eta_{t-1}$ and rearranging this expression yields,

$$
y_t = w_t + \frac{1 - \sqrt{q_{t-1}}}{1 + \sqrt{q_t}}\sqrt{\frac{\eta_t}{\eta_{t-1}}}(w_t - w_{t-1}),
$$

as claimed. $\square$

**Proposition 37.** Assume that $f$ is $\mu$-strongly convex and $\bar{L}$-smooth. Then the iterates $w_t$, $z_t$, and $y_t$ satisfy the following convergence rates,

$$
\begin{aligned}
f(w_T) - f(w_*) &\leq \left[\prod_{t=0}^{T-1}(1 - \sqrt{\eta_t \cdot \mu})\right]\left(f(w_0) - f(w_*) + \frac{\mu}{2}\|w_0 - w_*\|_2^2\right), \\
f(z_T) - f(w_*) &\leq \left(\frac{\bar{L}}{\mu}\right)\left[\prod_{t=0}^{T-1}(1 - \sqrt{\eta_t \cdot \mu})\right]\left(f(w_0) - f(w_*) + \frac{\mu}{2}\|w_0 - w_*\|_2^2\right), \qquad (30) \\
f(y_T) - f(w_*) &\leq \left(\frac{\bar{L}}{\mu}\right)\left[\prod_{t=0}^{T-1}(1 - \sqrt{\eta_t \cdot \mu})\right]\left(f(w_0) - f(w_*) + \frac{\mu}{2}\|w_0 - w_*\|_2^2\right).
\end{aligned}
$$

*Proof.* We follow the proof strategy developed by previous work (d'Aspremont et al., 2021). Since $f$ is $\bar{L}$-smooth, the backtracking line-search returns a step-size $\eta_t \geq 1/2\bar{L}$ satisfying the Armijo condition (Nocedal & Wright, 1999). Strong convexity of $f$, convexity of $f$, and the Armijo line-search condition eq. (18) now imply the following inequalities,

$$
\begin{aligned}
f(y_t) - f_* + \langle \nabla f(y_t), w_* - y_t \rangle + \frac{\mu}{2}\|w_* - y_t\|^2 &\leq 0 \\
f(y_t) - f(w_t) + \langle \nabla f(y_t), w_t - y_t \rangle &\leq 0 \\
f(w_{t+1}) - f(y_t) + \frac{\eta_t}{2}\|\nabla f(y_t)\|^2 &\leq 0.
\end{aligned}
$$

Recall $q_t = \eta_t \cdot \mu$; we have by Lemma 35 that $\eta_t \leq \frac{1}{\mu}$. If $\eta_t = 1/\mu$, then this lemma also implies $f(w_{t+1}) = f(w_*)$. On the other hand, if $\eta_t < 1/\mu$, then $q_t < 1$ and the following weighted sum of inequalities is valid,

$$
\begin{aligned}
\frac{\sqrt{q_t}}{1 - \sqrt{q_t}} &\left[ f(y_t) - f_* + \langle \nabla f(y_t), w_* - y_t \rangle + \frac{\mu}{2} \| w_* - y_t \|^2 \right] \\
&+ \left[ f(y_t) - f(w_t) + \langle \nabla f(y_t), w_t - y_t \rangle \right] \\
+ \frac{1}{1 - \sqrt{q_t}} &\left[ f(w_{t+1}) - f(y_t) + \frac{\eta_t}{2} \| \nabla f(y_t) \|^2 \right] \\
&\leq 0.
\end{aligned}
\tag{31}
$$

Using basic algebraic operations, it is possible to show that eq. (31) is equivalent to,

$$
f(w_{t+1}) - f(w_*) + \frac{\mu}{2} \| z_{t+1} - w_* \|^2 \leq (1 - \sqrt{q_t}) \left[ f(w_t) - f(w_*) + \frac{\mu}{2} \| z_t - w_* \|^2 \right].
\tag{32}
$$

For example, this can be established by subtracting eq. (31) from eq. (32), applying the update rules in eq. (17), and then reducing to monomials and cancelling terms. In keeping with past work (d'Aspremont et al., 2021), we omit the calculations and use Mathematica (Wolfram Research, Inc., 2024) to show the equivalence symbolically. See `reformulation_proof.nb` in the supplementary material for the associated code. Recursing on eq. (32) from $t = T$ to 0 shows,

$$
f(w_T) - f(w_*) + \frac{\mu}{2} \| z_T - w_* \|^2 \leq \left[ \prod_{t=0}^{T} (1 - \sqrt{\eta_t \cdot \mu}) \right] \left( f(w_0) - f(w_*) + \frac{\mu}{2} \| w_0 - w_* \|_2^2 \right),
\tag{33}
$$

from which we immediately deduce the claimed convergence rate on $f(w_T)$. The convergence rate for $z_T$ follows from,

$$
\begin{aligned}
f(z_T) - f(w_*) &\leq \frac{\bar{L}}{2} \| z_T - w_* \|^2 \\
&\leq \left( \frac{\bar{L}}{\mu} \right) \left[ \prod_{t=0}^{T-1} (1 - \sqrt{\eta_t \cdot \mu}) \right] \left( f(w_0) - f(w_*) + \frac{\mu}{2} \| w_0 - w_* \|_2^2 \right).
\end{aligned}
$$

Finally, since $y_T$ is a convex combination of $w_T$ and $z_T$, we deduce $f(y_T) \leq \max \{ f(w_T), f(z_T) \}$. This completes the proof. $\qquad\square$

*Proof of Theorem 21.* We follow the analysis structure outlined in Section 3.5:

1. Our region of interest under glocal smoothness is $\{ w \mid f(w) - f_* \leq \delta \}$.

2. By Proposition 37, we know that

$$
\begin{aligned}
\max \{ f(w_t), f(y_t) \} &\leq \left( \frac{L}{\mu} \right) \left[ \prod_{i=0}^{t-1} (1 - \sqrt{\eta_i \cdot \mu}) \right] \left( f(w_0) - f(w_*) + \frac{\mu}{2} \| w_0 - w_* \|_2^2 \right) \\
&=: h_t.
\end{aligned}
$$

   Let $t_\delta$ be the first $t$ for which $h_t \leq \delta$. Since $h_t$ is monotone decreasing in $t$, $h_t \leq h_{t_\delta} \leq \delta$ for all $t \geq t_\delta$. Thus, $f(w_t) - f_*, f(y_t) - f_* \leq \delta$ for all $t \geq t_\delta$.

3. Since $f$ is globally $L$-smooth, the step-size selected by backtracking line-search on eq. (18) satisfies $\eta_t \geq 1/2L$ for all $t \in \mathbb{N}$. As a result, the sequence $h_t$ is controlled as,

$$
h_t \leq \left( \frac{L}{\mu} \right) \left[ 1 - \sqrt{\frac{\mu}{2L}} \right]^t \left( f(w_0) - f(w_*) + \frac{\mu}{2} \| w_0 - w_* \|_2^2 \right).
$$

We conclude that an upper-bound on the iteration complexity for $w_t, y_t$ to enter and remain in the $\delta + f_*$ sub-level set is given by,

$$t_\delta \leq \left\lceil \sqrt{\frac{2L}{\mu}} \log \left( \frac{L \left( f(w_0) - f_* + \frac{\mu}{2} \|w_0 - w_*\|^2 \right)}{\mu \cdot \delta} \right) \right\rceil.$$

4. We have established that $f(y_t) - f_*, f(w_t) - f_* \leq \delta$ for all $t \geq t_\delta$. It only remains to show an improved bound on the step-sizes from line-search for all $t \geq t_\delta$ due to $L_*$-smoothness over the $\delta + f_*$ sub-level set. In particular, we shall show $\eta_t \geq \frac{1}{2L_*}$ for all $t \geq t_\delta$. Proposition 7 implies that for all $t \geq t_\delta$ and $\eta \leq 1/L_*$,

$$\tilde{w}_{t+1}(\eta) = y_t - \eta \nabla f(y_t),$$

satisfies $f(\tilde{w}_{t+1}(\eta)) - f_* \leq \delta$. Thus, $L_*$ smoothness holds between $\tilde{w}_{t+1}(\eta)$ and $y_t$, implying,

$$f(\tilde{w}_{t+1}(\eta)) \leq f(y_t) - \eta \left( 1 - \frac{\eta L_*}{2} \right) \|\nabla f(y_t)\|^2.$$

As a result, the Armijo condition eq. (18) is satisfied for all $\eta \leq 1/L_*$. The backtracking line-search with backtracking parameter $1/2$ is thus guaranteed to return $\eta_t \geq \frac{1}{2L_*}$ for all $t \geq t_\delta$. We conclude the following tighter upper-bound on $f(w_t) - f_*$ for $t \geq t_\delta$,

$$f(w_t) - f_* \leq \left[ 1 - \sqrt{\frac{\mu}{2L_*}} \right]^{t-t_\delta} \left[ 1 - \sqrt{\frac{\mu}{2L}} \right]^t \left( f(w_0) - f(w_*) + \frac{\mu}{2} \|w_0 - w_*\|_2^2 \right)$$

$$\leq \frac{\mu}{L} \left[ 1 - \sqrt{\frac{\mu}{2L_*}} \right]^{t-t_\delta} \delta.$$

5. Combining the two complexities implies we require at most

$$T \leq \sqrt{\frac{2L}{\mu}} \log \left( \frac{L \left( f(w_0) - f_* + \frac{\mu}{2} \|w_0 - w_*\|^2 \right)}{\mu \cdot \delta} \right) + \sqrt{\frac{2L_*}{\mu}} \log \left( \frac{\mu \cdot \delta}{L \cdot \epsilon} \right),$$

iterations to compute an $\epsilon$ sub-optimal point.

$\square$

## K  Non-Linear Conjugate Gradient Convergence Results

Classic non-linear conjugate gradient (NLCG) methods use iterations of the form

$$w_{t+1} = w_t + \eta_t(-\nabla f(w_t) + \beta_t(w_t - w_{t-1})), \tag{34}$$

and typically differ in their choice of $\beta_t$. A common variant is the Polak-Ribiére-Polyak (PRP) choice (Polak & Ribiere, 1969; Polyak, 1969) of

$$\beta_t = \frac{\|\nabla f(w_t)\|^2 - \nabla f(w_t)^T \nabla f(w_{t-1})}{\|\nabla f(w_{t-1})\|^2}, \tag{35}$$

where we may periodically reset the algorithm by setting $\beta_t = 0$. It is well known that NLCG methods with LO will terminate in at most $d$ steps when applied to a $d$-dimensional strongly-convex quadratic function (Nocedal & Wright, 1999). Further, non-linear conjugate gradient methods typically outperform gradient descent methods in practice by a significant margin even for non-quadratic functions. However, for non-quadratic functions the global iteration complexity of NLCG is worse than for gradient descent (Das Gupta et al., 2024). In this section we use a glocal assumption to present an improved iteration complexity for NLCG that justifies its use in cases where the Hessian matrix is well-behaved in the local region.

In particular, we consider the extreme case where the Hessian is constant in the local region. This implies that the function is locally quadratic near the solution. Examples of functions that can be locally quadratic around the solution include the Huber loss (Hastie et al., 2017) as well as smooth variants on support vector machines (Lee & Mangasarian, 2001; Rosset & Zhu, 2007). In this setting, our convergence result exploits a recent global analysis of the PRP choice (Das Gupta et al., 2024) and the classic argument that NLCG methods converge at an accelerated rate and in at most $d$ steps if restarted on a locally-quadratic function (Nocedal & Wright, 1999). Our analysis relies on the following recent result regarding the performance of the PRP variant of NLCG (Das Gupta et al., 2024).

**Lemma 38.** Assume that $f$ is $\bar{L}$-smooth and $\mu$-strongly convex. For all $t$ the iterates of NLCG as defined in (34) with step-size $\eta_t$ given by line optimization and $\beta_t$ given by the PRP momentum rate (35) the guaranteed progress is

$$f(w_t) - f_* \leq \left( \frac{1 - (\mu/L)^2}{1 + (\mu/L)^2} \right)^2 (f(w_{t-1}) - f_*).$$

Note that this result assumes that the PRP update is performed at all iterations, but it implies that the same rate holds for variants with resets (since a reset at iteration $t$ would correspond to initializing the PRP method at $w_t$). Besides monotonic decrease of $f(w_t) - f_*$, the above result implies that if we start from $\bar{w}_0$ we require $t = (1/2)\frac{\bar{L}^2}{\mu^2} \log\left( \frac{f(\bar{w}_0) - f_*}{\bar{\epsilon}} \right)$ iterations to have $f(w_t) - f_* \leq \bar{\epsilon}$.

To bound the number of iterations in the local region after the first reset, we also require two classic results regarding the convergence of the linear conjugate gradient method applied to quadratic functions (Polyak, 1987, Section 3.2.2)

**Lemma 39.** Assume that $f$ is $\bar{L}$-smooth and $\mu$-strongly convex quadratic function,

$$f(w) = \frac{1}{2}w^T A w + b^T w + \gamma, \tag{36}$$

for a positive-definite matrix $A$, vector $b$, and scalar $\gamma$. Consider the linear conjugate gradient method

$$w_{k+1} = w_k - \eta_k \nabla f(w_k) + \beta_k(w_k - w_{k-1}),$$

where $\eta_k$ and $\beta_k$ are jointly set to minimize $f$. The algorithm terminates with the solution in at most $d$ iterations and beginning from $\bar{w}_0$ also satisfies

$$\|w_t - w_*\| \leq 2 \left( \frac{\bar{L}}{\mu} \right)^{1/2} \left( \frac{1 - \sqrt{\mu/\bar{L}}}{1 + \sqrt{\mu/\bar{L}}} \right)^t \|\bar{w}_0 - w_*\|.$$

Note that by using

$$\frac{\mu}{2}\|w - w_*\|^2 \leq f(w) - f_* \leq \frac{L}{2}\|w - w_*\|^2,$$

we can use the rate above to bound the progress in sub-optimality,

$$f(w_t) - f_* \leq 4 \left( \frac{\bar{L}}{\mu} \right)^2 \left( \frac{1 - \sqrt{\mu/\bar{L}}}{1 + \sqrt{\mu/\bar{L}}} \right)^{2t} (f(\bar{w}_0) - f_*).$$

The relevance of this result to our setting is that functions with a constant Hessian can be written in the form (36), and that the PRP NLCG method with exact line optimization to set $\eta_k$ is equivalent to the linear conjugate gradient update when applied to quadratic functions. This implies that for quadratic functions we require $t = (1/2)\sqrt{L/\mu}\log(4(L/\mu)^2(f(\bar{w}_0) - f_*)/\bar{\epsilon})$ iterations to have $f(w_t) - f_* \leq \bar{\epsilon}$ if PRP NLCG is initialized at $\bar{w}_0$. Given the above results, we now present our main convergence theorem for NLCG.

**Theorem 40.** Assume that $f$ is glocally $(L, L_*, \delta)$-smooth, $\mu$-strongly convex, and the Hessian $\nabla^2 f(w)$ is constant over the region where $f(w) - f_* \leq \delta$. For all $t \geq 0$, let $w_t$ be the iterates of NLCG as defined in (34) with step-size $\eta_t$ given by line optimization. Set $\beta_t$ to 0 on the first and every $d$th iteration, and $\beta_t$ set using the PRP formula (35) on all other iterations. Then $f(w_T) - f_* \leq \epsilon$ for all

$$T \geq O\left(\frac{L^2}{\mu^2} \log\left(\frac{\Delta_0}{\delta}\right)\right)$$

$$+ \min\left\{ O\left(\frac{L^2}{\mu^2} \log\left(\frac{\delta}{\epsilon}\right)\right), d + \min\left\{ d, O\left(\sqrt{\frac{L_*}{\mu}} \log\left(\left(\frac{L_*}{\mu}\right)^2 \frac{\delta}{\epsilon}\right)\right)\right\}\right\}.$$

*Proof of Theorem 40.* We follow the analysis structure outlined in Section 3.5:

1. Our region of interest under glocal smoothness and is $\{w \mid f(w) - f_* \leq \delta\}$.

2. Let $t_\delta$ be any iteration where $f(w_{t_\delta}) - f_* \leq \delta$. From the monotonicity in Lemma 38, it follows that $w_t$ stays in the region of interest for subsequent $t$.

3. From Lemma 38, PRP NLCG initialized at $w_0$ is guaranteed to satisfy $f(w_t) - f_* \leq \delta$ after $t_\delta = \left\lceil \frac{1}{2}\left(\frac{L}{\mu}\right)^2 \log\left(\frac{f(w_0) - f_*}{\delta}\right)\right\rceil$ iterations.

Within the local region, there are 3 different ways we could reach an accuracy of $\epsilon$: (a) we might reach an accuracy of $\epsilon$ before the first reset occurs due to the global rate from Lemma 38, (b) we might reach an accuracy of $\epsilon$ after the first reset due to the accelerated local rate from Lemma 39, or (c) we might reach an accuracy of $\epsilon$ due to the finite termination in at most $d$ steps after the first reset from Lemma 39. We consider case (a) first:

4. Initialized at $w_{t_\delta}$, from Lemma 38 we require an additional $t = \left\lceil \frac{1}{2}\left(\frac{L}{\mu}\right)^2 \log\left(\frac{f(w_{t_\delta}) - f_*}{\epsilon}\right)\right\rceil$ iterations to have $f(w_t) - f_* \leq \epsilon$.

5. Using that $f(w_{t_\delta}) \leq \delta$, it is sufficient to have $t = \left\lceil \frac{1}{2}\left(\frac{L}{\mu}\right)^2 \log\left(\frac{\delta}{\epsilon}\right)\right\rceil$ iterations.

We now turn to case (b):

4. In order to exploit the local quadratic property, we require at most $(d - 1)$ additional iterations beyond $t_\delta$. Thus, after $t_\delta + (d - 1)$ iterations the algorithm is in in the quadratic region and has already reset so NLCG is equivalent to linear CG. Using Lemma 39 with $\bar{L} = L_*$, PRP NLCG is guaranteed to satisfy $f(w_t) - f_* \leq \epsilon$ after another $t = \left\lceil \frac{1}{2}\sqrt{\frac{L_*}{\mu}} \log\left(4\left(\frac{L_*}{\mu}\right)^2 \frac{f(w_{t_\delta + (d-1)}) - f_*}{\epsilon}\right)\right\rceil$ iterations.

5. Using that $f(w_{t_\delta}) - f_* \leq \delta$ and Lemma 38, we have $f(w_{t_\delta + (d-1)}) - f_* \leq \left(\frac{1 - (\mu/L)^2}{1 + (\mu/L)^2}\right)^{2(d-1)} \delta \leq \exp(-2(d - 1)\mu^2/L^2)\delta$, we can conclude that it is sufficient to have $t = \left\lceil \frac{1}{2}\sqrt{\frac{L_*}{\mu}} \log\left(4\left(\frac{L_*}{\mu}\right)^2 \frac{\delta}{\epsilon}\right) - (d-1)\sqrt{\frac{L_*}{\mu}}\frac{\mu^2}{L^2}\right\rceil$ additional iterations (the negative term reduces the iteration count based on the progress made before the reset, which we have omitted in the final result, though it could be significant if $d$ is large and the condition number $L/\mu$ is small).

Finally, in case (c) we require at most $(d - 1)$ steps before the first reset and then at most $d$ additional steps to solve the problem due to the finite termination in Lemma 39. The final result is the minimum among cases (a), (b), and (c). $\square$

The first term in the iteration complexity is the number of iteration to reach the local region. The squared dependency on $(L/\mu)$ reflects the potential slow global convergence rate of NLCG. The first term in the outer minimum reflects that the global convergence rate may lead to an accuracy of $\epsilon$ before the first reset occurs in the local region. The $d$ in the outer minimum bounds the number of iterations performed in the local region before a restart has occurred. The $d$ in the inner minimum reflects that NLCG will converge in exactly $d$ steps after being restarted in the local region. The other term in the inner minimum reflects the post-reset accelerated rate of NLCG with local smoothness $L_*$ within the local region (the squared $L_*/\mu$ term in the log is due to bounding the function values using a local iterate-based analysis of conjugate gradient). Note that this accelerated rate does not require knowing $\mu$. Based on this result, we expect NLCG to outperform other first order methods for sufficiently small values of $d$, provided that $\Delta_0/\delta$ and the global condition number $(L/\mu)$ are not too large.

## L   Additional Lemmas

We use the following well-known result (Bubeck et al., 2015, Lemma 5) in some of our proofs.

**Lemma 41.** Let $f$ be an $L$-smooth and convex function. Then $f$ satisfies

$$f(y) - f(x) \leq \langle \nabla f(y), y - x \rangle - \frac{1}{2L} \|\nabla f(y) - \nabla f(x)\|^2.$$

The next result (Drori, 2018, Theorem 3.1) shows that Lemma 41 also applies over open sets.

**Lemma 42.** Let $f : C \to \mathbb{R}$ be an $L$-smooth and convex function over an open set $C \subsetneq \mathbb{R}^d$. Then for any $x, y \in C$ such that $\|x - y\| < \text{dist}(y, \mathbb{R}^d \setminus C)$,

$$f(y) - f(x) \leq \langle \nabla f(y), y - x \rangle - \frac{1}{2L} \|\nabla f(y) - \nabla f(x)\|^2.$$

The following lemma (Beck & Tetruashvili, 2013, Lemma 3.5) is used in our convex results.

**Lemma 43.** Let $\{A_k\}_{k \geq 0}$ be a non-negative sequence of real numbers satisfying

$$A_k - A_{k+1} \geq \gamma A_k^2,$$

for $k = 0, 1, \ldots$, and

$$A_0 \leq \frac{1}{m\gamma},$$

for positive $\gamma$ and $m$. Then

$$A_k \leq \frac{1}{\gamma} \frac{1}{k + m},$$

for $k = 0, 1, \ldots$.

## M   Open Problems

In this section we discuss a variety of possible follow-up works and related open problems.

### M.1   Analyzing LO

The Armijo condition is a standard approach to rule out step sizes that are too large. We use the Armijo condition in our results related to SGD (Theorem 20) and NAG (Theorem 21). It would be preferable to have an analysis of LO in these settings (or prove that LO does not achieve the appropriate rate). LO can allow step sizes that are much larger than allowed by the Armijo condition, but does not guarantee the same sufficient decrease relative to the step size. In particular, we did not find results in the literature (positive or negative) related to showing accelerated rates for NAG(LO) or analyzing the convergence of SGD with LO under interpolation. It is known that the empirical performance of SGD under interpolation can be improved with step sizes that are larger than allowed by the Armijo condition (Galli et al., 2023). A related issue is whether $R^2(w_0)$ can be replaced by $\|w_0 - w_*\|^2$ when only assuming global convexity (Theorems 14 and 31) so that the iteration complexity GD(LO) is never worse than GD(1/L).

## M.2 Adapting NAG to $\mu_*$

In our analysis of NAG (Theorem 21) we assume that $\mu$ is given. This could potentially be replaced with a restart mechanism (Nesterov, 2013). However, similar to GD methods we would like NAG to be adaptive to a local $\mu_*$. We view it as an open problem to develop an NAG method that can adapt to a local $L_*$ and $\mu_*$.

## M.3 Relaxing interpolation

Our analysis of SGD (Theorem 20) exploits deterministic monotonicity in the iterate distance under the interpolation assumption. Without interpolation the first challenge of analyzing SGD under glocal smoothness is that SGD may leave the local $\delta$ region repeatedly. To analyze SGD without interpolation it might be easier to use a continuous variant of the glocal smoothness assumption (Vaswani & Babanezhad, 2025), use glocal assumptions on the noise as in the state-dependent noise assumptions (Ilandarideva et al., 2024), or focus on variants such as dual averaging (Nesterov, 2009) and finite-sum methods (Schmidt et al., 2017) where the variance in the stochastic gradients converges to zero.

A second challenge when using SGD under glocal-like assumptions is finding a strategy for setting the step sizes which adapts to local properties. Using a decoupled variant of the stochastic line-search in conjunction with exponentially decreasing step-sizes (Vaswani et al., 2022), or the stochastic Polyak step-size with Ada-Grad step-sizes (Jiang & Stich, 2023) might be promising approaches that can take advantage of the glocal assumption, while ensuring convergence to the minimizer.

## M.4 Improving NLCG

It is possible to relax the assumption that the Hessian is constant in the local region when analyzing NLCG. In particular, if the Hessian is only Lipschitz continuous then the $d$-iteration cycles of NLCG have an asymptotic quadratic local convergence rate (Polyak, 1969). It is possible that a glocal analysis could incorporate this property. It may also be possible to improve the global rate under a variant of NLCG that exploits subspace optimization (Karimi & Vavasis, 2017).

## M.5 Generalizing to PL functions

In Section 5.4, we mention that our glocal GD(LO) result for strongly-convex functions easily generalizes to PL functions. We note that our coordinate descent bounds also still hold if the PL condition is assumed instead of strong convexity. However, many of our results do not immediately apply to general PL functions. In particular, we use other properties of strong convexity for our bounds related to the Polyak step size, AdGD step size, SGD, accelerated GD, and NLCG. Thus, showing improved convergence rates under these settings for PL remains open.

## M.6 Non-convexity

This work focuses on functions that are either convex or satisfy the PL inequality. It is likely that glocal smoothness could be combined with related conditions like gradient-dominated and star-convex functions (Nesterov & Polyak, 2006). However, it is not clear how to exploit a condition like glocal smoothness for general non-convex functions. The challenge in the general non-convex setting is bounding the time before we can guarantee that we have a sub-optimality below $\delta$. One possibility is to define the $\delta$ region based on the norm of the gradient. However, it is unclear (except asymptotically) how we can guarantee that an iterate reaches/stays in a region with small gradient norm.

## M.7 Non-monotone line-search

There is a long history of line-search methods that do not require monotonic decrease in the objective (Grippo et al., 1986; Raydan, 1997), and recent work shows that such methods can be effective in training over-parameterized neural networks (Galli et al., 2023). It would be interesting to analyze such schemes under

glocal smoothness, and to explore additional assumptions under which these practically-effective methods have a theoretical advantage.

## M.8 Adaptive non-uniform sampling

Our analysis of coordinate descent (Theorem 17) and SGD under interpolation (Theorem 20) assume uniform sampling of the coordinates and examples (respectively). However, it is known that the global convergence rates in these settings can be improved using a non-uniform distribution incorporating the Lipschitz constants of the coordinates/examples (Nesterov, 2012; Needell et al., 2014). It is also known that faster theoretical rates can be obtained using known local Lipschitz constants (Vainsencher et al., 2015), and faster rates in practice are reported when estimating local constants (Schmidt et al., 2017). However, we are not aware of theoretical results on non-uniform sampling based on estimates of local constants (which is challenging because the estimate of smoothness for each coordinate/example affects the sampling distribution).

## M.9 Per-coordinate step sizes

A key component of the Adam optimizer (Kingma & Ba, 2014) is a per-variable learning rate. Recent works have given the first improved convergence rates with appropriately-estimated per-variable learning rates under global smoothness (Kunstner et al., 2023; Gao et al., 2024). It is not obvious what the analogue of LO should be in this setting. In the glocal setting, we note that prior works may not adapt quickly enough in the $\delta$ region (Kunstner et al., 2023) or quickly enough globally (Gao et al., 2024). Thus, we expect that better methods could be developed under glocal smoothness.

## M.10 Level sets

We have considered partitioning the space into a global region and a sub-level set, but this is not always the most appropriate partition. For example, using the Huber loss for regression (Hastie et al., 2017) the smoothness constant can be maximized at the minimizer. There is also empirical evidence that the local smoothness constant can increase during the training of deep neural networks (Cohen et al., 2021; Tran et al., 2024). For such settings it could make sense to assume global $L$-smoothness and $L_*$-smoothness over a superlevel set. We chose to focus on sublevel sets in our glocal assumption since good performance requires increasing the step size, whereas an increased smoothness over a superlevel set is already exploitable by simple backtracking procedures. Perhaps a more general assumption would be that the function is $L_*$-smooth for a particular unknown range of function values (whether it be a sub/superlevel set or not), and we would like to optimize iteration complexity in this setting.

## M.11 Non-smooth functions

We have focused on the case of unconstrained smooth functions, but we expect that it is possible to analyze algorithms for constrained and non-smooth optimization under similar assumptions. For example, non-smooth optimization typically relies on sub-gradient bounds. A non-smooth variant of the glocal assumption could assume that the sub-gradient bound is smaller on a sublevel set. Indeed, existing work (Diakonikolas & Guzmán, 2024) shows improved rates based on the sub-gradients near the solution. It would be interesting to explore whether the adaptivity of AdaGrad (Duchi et al., 2011) is preserved if the subgradients become more well-behaved over time, or if an alternate algorithm is required.

## M.12 Relative smoothness

We expect that it would be possible to analyze algorithms under glocal variants of relative smoothness (Bauschke et al., 2017; Lu et al., 2018) and related assumptions (Maddison et al., 2018; Wilson et al., 2019). The modified algorithm obtained for several canonical problems under these relaxed assumptions is GD with a different step size sequence, and it would be interesting to know if GD(LO) obtains faster rates on these problems.

### M.13 Neural network training

Our assumptions of convexity and glocal smoothness are meant to be the simplest possible assumptions showing the benefits of a global-local analysis, but do not hold for typical neural networks. However, there are potential scenarios where a similar two phase style analysis could make sense. For example, as discussed in Section 5.4 several works argue that the PL inequality may hold around the optimization trajectory during neural network training. It is plausible that glocal smoothness assumptions may also hold around the optimization trajectory. Indeed, the work by Gilmer et al. (2021) shows that step size warm-up can induce a decrease in the local smoothness constant. Thus, the iterates early in training can be in a region with a larger Lipschitz constant while the iterates of the post warm-up phase enter a region with a smaller local Lipschitz constant.

We would also like to comment on the edge of stability phenomenon (Cohen et al., 2021) and other related phenomena. The edge of stability phenomenon suggests, in several deep learning training scenarios, that the smoothness of the objective adapts to the step size being used. This is as opposed to the present work, where we are advocating to adapt the step size to the problem. Nevertheless, it is likely that algorithmically controlling local smoothness is important for improving the training of deep neural networks. Indeed, sharpness-aware minimization methods (Foret et al., 2021) push iterates into regions with lower smoothness constants as training progresses and have been shown to improve generalization. Thus, we believe that improving our understanding of how local smoothness interacts with optimization may ultimately shed light on how to better train deep neural networks.

