# OpenReview forum: "Glocal Smoothness: Line search and adaptive sizes can help in theory too!"
_TMLR — Accepted by TMLR_

### Review · Reviewer_Coac · 2026-03-01

**Summary Of Contributions:**

The paper introduces a notion of glocal smoothness, where a function is $L$-smooth globally and $L^\*$-smooth locally (on the sublevel set).
This assumption depends only only the properties of the function, and does not depend on the iterates of the algorithm, which makes it possible to compare the iteration complexities of different methods.
Under $\mu$ strong convexity, the authors prove that gradient descent with line optimization, GD(LO), has iteration complexity
$T \ge \frac{L}{\mu} \log \left(\frac{\Delta_0}{\delta}\right) + \frac{L^\*}{\mu} \log \left(\frac{\delta}{\epsilon}\right)$.
This improves over the standard $ \frac{L}{\mu} \log \left(\frac{\Delta_0}{\epsilon}\right)$ rate when $L^\* < L$.
They also show that GD(LO) is faster than NAG with step size $1/L$ under local smoothness.
A key technical result shows that once an iterate enters the $\delta$-sublevel set, steps with size controlled by $1/L^\*$ remain in this region, enabling clean two-phase analyses.
The framework is extended to adaptive step sizes, coordinate optimization, SGD methods, NAG, and nonlinear conjugate gradient methods.

**Audience:**

Yes

**Audience Explanation:**

The paper provides a simple assumption that enables direct comparison between line search and acceleration, and extends to several widely used methods. This is well aligned with the interests of a theoretical machine learning audience.
The list of open problems in appendix M is very interesting and shows that the proposed glocal condition is relevant and could potentially improve results in in many lines of work in optimization theory.
Additionally, the paper is very readable. The definition of glocal smoothness is simple, the main bound (theorem 6) is clean, and the proof sketch has a readable form.

**Claims And Evidence:**

Yes

**Claims Explanation:**

The main theoretical claims are supported by clear statements, structured proofs, and explicit assumptions.
The two-phase complexity results follow cleanly from strong convexity (or PL) and the glocal assumption, and the lemma ensures that iterates remain in the local region once entered.
The comparison to fixed-step NAG is formalized through an explicit condition.
The paper explicitly notes that exact line optimization is often impractical and discusses Armijo-style methods and other adaptive rules as more realistic alternatives.
Empirical evidence is limited, so the support is strongest on the theoretical side rather than experimental validation.

**Requested Changes:**

I would like to ask for a bit more explanations on the following two points:
- Many key results rely on global smoothness and either strong convexity or the PL inequality. While these assumptions are arguably standard in this community, it would be helpful to comment on how well they hold in modern architectures.
- The proof for Lemma 3 (appendix B) seems to drop the factor 1/4 somewhere for the local smoothness bound.

---

> ### Author Response · Authors · 2026-03-18
> **Response to Reviewer Coac**
>
> Thank you for taking the time to read and comment on the work. Thank you also for highlighting the clarity/cleanliness
> of the results and the alignment with the interests of a theoretical ML audience. We appreciate the comments regarding
> the open problems section, and do think that the work will lead to improvements in various aspects of optimization theory.
> Below we comment on the two requested changes.
>
> We note that our glocal assumption is meant to be the simplest possible assumption showing the benefits of a global-local
> analysis, but it is not quite the right assumption for neural networks. However, there are situations where a similar two-phase
> style analysis could make sense and we hope that these ideas will ultimately help with neural network training. For example,
> the work by Gilmer et al. [2021] shows that the popular step size warm-up strategy can induce a decrease in sharpness, so
> one could think of the iterates early in training as being in a region with a larger Lipschitz constant while the iterates of the
> post warm-up phase enter a region with a smaller, local Lipschitz constant. Similarly, sharpness-aware minimization methods
> [Foret et al., 2021] also push iterates into regions of lower sharpness as training progresses. For the PL inequality, in the
> paper we highlight the work by Liu et al. [2022] that shows that a variant of PL holds for over-parameterized networks. Again
> in the context of neural networks, a more recent paper shows that a local variant of the PL inequality also holds in locally
> quasi-convex regions satisfying a local neural tangent kernel (NTK) stability assumption [Aich et al., 2026]. The paper does
> already very-briefly discuss neural networks in the edge of stability section, but it is a good idea to expand on this point so
> we plan to add more discussion of connections/relevance to neural networks to the paper.
>
> The proof of Lemma 3 does not use the 1/4 factor in the local smoothness bound, and this was accidentally pasted into
> the statement of the result. Thanks for catching this.
>
> Due to time constraints and other work commitments, we haven’t been able to fully update the paper yet. However, we
> will upload the updated paper with the requested changes as soon as we can and provide the list of changes once we have
> done so.
>
> **References**
>
> A. Aich, A. B. Aich, and B. Wade. From sublinear to linear: Fast convergence in deep networks via locally polyak-lojasiewicz
> regions. CoRR, abs/2507.21429, 2025.
>
> P. Foret, A. Kleiner, H. Mobahi, and B. Neyshabur. Sharpness-aware minimization for efficiently improving generalization.
> In International Conference on Learning Representations, 2021.
>
> J. Gilmer, B. Ghorbani, A. Garg, S. Kudugunta, B. Neyshabur, D. Cardoze, G. E. Dahl, Z. Nado, and O. Firat. A loss
> curvature perspective on training instability in deep learning. CoRR, abs/2110.04369, 2021.
>
> C. Liu, L. Zhu, and M. Belkin. Loss landscapes and optimization in over-parameterized non-linear systems and neural
> networks. Applied and Computational Harmonic Analysis, 59:85–116, 2022.

---

> ### Author Response · Authors · 2026-03-30
> **Response to Coac with Paper Changes**
>
> We have now uploaded the updated version of our paper, addressing your requested changes as follows:
> 1. In Section 5.4 and Section M.13, we have expanded our discussion of the relevance of the glocal and PL assumptions
> to neural network training, incorporating the references given in our previous response.
> 2. In Section 3.1, we have modified the Lemma 3 result and removed the extra factor of 1/4.
>
> Note that all non-trivial changes in the paper are highlighted in red.

---

### Review · Reviewer_ERFL · 2026-03-07

**Summary Of Contributions:**

As the paper itself points out, the basic idea of dividing space into sub-regions with different assumptions that allow achieving tighter complexity bounds is not new. But the framing presented in this paper makes many results elegant and intuitive, which is the most important contribution of this paper. Devising results which are on the one hand elegant, while on the other hand being useful is an art, and any paper succeeding in doing so is worthy of publication and contributes to the community and dissemination of knowledge.

I do believe that several things need to be revised, reframed, or even improved before the paper is ready for publication, for example:
1. The intro claims about theory predicting an error bound as a function of the number of iterations, but forgets the constants. For example, NAG's constant is $2 L || x_0 - x^{opt} ||^2$, whereas GD's constant is $(L/2) || x_0 - x^{opt} ||^2$. Therefore, even the theoretical bound of NAG is not always better than the one of GD. Additionally, bounds are just bounds - they aren't wrong, they can simply be loose. The theory doesn't claim something that doesn't happen in practice.  The claims in the intro need to be made more precise and rigorous.
2. There is occasionally a feeling that this paper does "half-the-job" in some areas, and under-utilizes the fact that the bounds are obtained _simultaneously for all $\delta$_. Since there is typically a dependency of $L^{\ast}$ on $\delta$, we could try and minimize the bound over $\delta$. The paper  provides an explicit formula for logistic regression, but doesn't "finish the job" in this sense.
3. Some literature is missing. Relative smoothness has been introduced in 2017, simultanously, by Bauschke et. al [1] in **2017** and then by Lu et. al. [2] in **2018**, but these papers aren't properly cited. Instead, a much later paper by Nesterov is cited.

[1]: Bauschke, Heinz H., Jérôme Bolte, and Marc Teboulle. "A descent lemma beyond Lipschitz gradient continuity: first-order methods revisited and applications." Mathematics of Operations Research 42.2 (2017): 330-348.
[2]: Lu, Haihao, Robert M. Freund, and Yurii Nesterov. "Relatively smooth convex optimization by first-order methods, and applications." SIAM Journal on Optimization 28.1 (2018): 333-354.

**Additional Comments:**

A very neat paper! I value simplicity in theory papers.

**Audience:**

Yes

**Audience Explanation:**

Citing "Devising results which are on the one hand elegant, while on the other hand being useful is an art, and any paper succeeding in doing so is worthy of publication and contributes to the community and dissemination of knowledge." from my review above.

**Claims And Evidence:**

No

**Claims Explanation:**

Intro requires some revision to be accurate, but can be made so.

**Requested Changes:**

1. Revise references. The two mentioned above  drew my attention immediately since they are from my field of expertise, but if these two references were missing or inaccurate, there probably are others.
2. Revise the intro to make the thesis of the paper more accurate, rigorous, and convincing.

---

> ### Author Response · Authors · 2026-03-18
> **Response to Reviewer ERFL**
>
> Thank you for taking the time to read and comment on the work. We especially appreciate the comment that the results are elegant and useful, which is our goal with this work. Below we comment on the proposed revisions.
>
> The introduction could indeed be more accurate on the points that we are discussing upper bounds, and that we are implicitly worried about the dependence on large values of variables (like $L$ and $1/\epsilon$) rather than constant factors. But we can and should make this more clear, and indeed such writing is one of the motivations for writing this paper in the first place; many works/researchers discount things like line search because "they do not improve the convergence rate''. We wanted to provide a framework under which line search leads to better upper bounds (and where the upper bounds may be closer to practical performance).
>
> It is a great point that we should place more emphasis on the fact that the bound holds for all $\delta$, and indeed that if you make $L_*$ a function of $\delta$ that we could take the best upper bound across $\delta$ values. We will update the paper to make this clear right away in the introduction, as well as after the first glocal result is presented. A concurrent work [Vaswani and
> Babanezhad, 2025] considered a related assumption that is like a continuous version of the glocal assumption (in $\delta$), and this led to a faster convergence rate for logistic regression (from sublinear to linear). On the other hand, logistic regression is quite special and for many problems we do not have simple ways to relate $L_*$ and $\delta$.
>
> For relative smoothness, we were intending to cite the Lu et al. paper (where Nesterov is a co-author) and it was simply an error to cite the Nesterov presentation. We will fix this and add the Bauschke et al. reference. We will also review the other references in the open problem section, but note that citations in the section are intended to be starting points on topics rather than comprehensive reviews of the literature. For the ideas that are most related to the main topic of the paper, we believe that Section 2 gives a fairly careful/comprehensive review/categorization.
>
> Due to time constraints and other work commitments, we haven't been able to fully update the paper yet. However, we will upload the updated paper with the requested changes as soon as we can and provide the list of changes once we have done so.
>
> **References**
>
> S. Vaswani and R. Babanezhad. Armijo line-search can make (stochastic) gradient descent provably faster. In International
> Conference on Machine Learning, 2025.

---

> ### Author Response · Authors · 2026-03-30
> **Response to ERFL with Paper Changes**
>
> We have now uploaded the updated version of our paper, addressing your requested changes as follows:
> 1. We have changed the text in Section 1 significantly to be more precise. We have also updated Section 3.5 and the
> conclusion based on the comments in the review.
> 2. We highlight at the end of Section 1 how the bounds hold for all $\delta$, and have added a new section (Section 3.7) that
> optimizes the bound over $\delta$ for logistic regression.
> 3. In Section M.12, we have added the 2 new relative smooth references as suggested.
>
> Note that all non-trivial changes in the paper are highlighted in red.

---

> > ### Comment · Reviewer_ERFL · 2026-03-30
> > **Much better!**
> >
> > Thank you for the revision. The paper now appears much more complete.

---

### Review · Reviewer_sQg4 · 2026-03-09

**Summary Of Contributions:**

The manuscript introduces the concept of glocal smoothness, a notion that bridges the gap between global and local smoothness. Specifically, a function is defined as glocally smooth if it is globally L-smooth while exhibiting a smaller smoothness constant within a sublevel set where the suboptimality is less than or equal to a given threshold.

By defining local smoothness independently of the specific iterates generated by an algorithm, the authors provide a framework that enables a direct comparison of the iteration complexities of different optimization methods. The authors show that the gradient descent with line search achieves faster theoretical convergence rates than the gradient descent with a fixed step size. The paper also studies how the glocal framework applies to practical step-size selection methods (such as Armijo, Polyak, and AdGD) as well as to other algorithms, including coordinate descent, stochastic gradient descent, and nonlinear conjugate gradient.
Overall, the paper is interesting and may be considered for publication after addressing the comments.

**Audience:**

Yes

**Audience Explanation:**

The paper introduces the notion of glocal smoothness, which bridges the gap between global and local smoothness assumptions in optimization. This concept provides a useful framework for analyzing optimization algorithms by distinguishing between the global smoothness of a function and a smaller smoothness constant within relevant sublevel sets. Such a perspective can help better explain the practical behavior of algorithms.

**Broader Impact Concerns:**

No significant broader impact concerns are apparent for this work. The paper is theoretical, focusing on the analysis of optimization algorithms.

**Claims And Evidence:**

Yes

**Claims Explanation:**

The claims are supported by clear and rigorous theoretical analysis. The assumptions are explicitly stated, and the results are justified through logical mathematical arguments and proofs.

**Requested Changes:**

Major Comments:

i) Definition 11. The authors should clarify whether the solution in this definition is assumed to be unique or whether the solution set may contain multiple elements.

ii) In the convex setting, local strong convexity appears to imply uniqueness of the solution. It would be helpful if the authors could either provide an example where the solution set is not a singleton under this assumption or revise the statement accordingly.

iii) In the presence of the PL condition, it seems that the results could potentially be extended to certain nonconvex problems. It would be interesting if the authors could discuss this point.

Minor Comment:

i) It would be preferable to present Equation (18) in Theorem 9 or earlier in the main text rather than placing it in the appendix.

---

> ### Author Response · Authors · 2026-03-18
> **Response to Reviewer sQg4**
>
> Thank you for taking the time to read and comment on the work. These requested changes will help to clarify various points, and we briefly comment on them below.
>
> We were implicitly assuming strong convexity when introducing Definition 11, implying a unique solution. The logical generalization for the convex case would be defining the local region based on the projection of $w$ onto the solution set. But in any case we will clarify the issue of uniqueness in this section.
>
> Local strong convexity indeed implies uniqueness of the solution, while the PL result and locally convex result (Appendix G) allow multiple solutions. Thanks for highlighting this issue, as it is a point we should emphasize.
>
> It is true that some of our results easily extend to non-convex problems satisfying the PL inequality, including the GD(LO) and coordinate descent results. However, it is more difficult to extend several of the results to the PL case. We will add a discussion regarding which results extend to general PL problems and which do not to the open problems section of our Appendix. We also plan to expand our discussion of non-convex functions in light of the comments of Reviewer Coac. Unfortunately, the general non-convex analysis seems challenging and we discuss this in the open problems section (Appendix M).
>
> Given its relevance to the practical step size section, it is a good point that the Armijo condition should be moved to the main paper.
>
> Due to time constraints and other work commitments, we haven't been able to fully update the paper yet. However, we will upload the updated paper with the requested changes as soon as we can and provide the list of changes once we have done so.

---

> ### Author Response · Authors · 2026-03-30
> **Response to sQg4 with Paper Changes**
>
> We have now uploaded the updated version of our paper, addressing your requested changes as follows:
> 1. In Section 5.1, we have now modified the definition to include cases where the solution is non unique, and clarified that
> $w_∗$ is unique in Definition 11 under strong convexity.
> 2. In Sections 5.3-4, we clarify that the solution is unique in the locally strongly convex case, but may not be unique in
> the locally-convex and PL cases.
> 3. In Section M.5, we now clarify for which of our results the strong convexity assumption can be easily replaced with the
> PL condition, and which results rely on other properties of strong convexity.
> 4. In Section 4, we now present the Armijo line search condition (in the main paper, as opposed to the Appendix
> previously).
>
> Note that all non-trivial changes in the paper are highlighted in red.

---

### Decision · Action_Editor_rNSL · 2026-04-14

**Recommendation:** Accept as is

**Audience:**

Yes

**Audience Explanation:**

Reviewers agree that the paper meets the criterion related to "Audience", and some even believe that it goes further by making a meaningful contribution more broadly to the analysis of optimisation algorithms.

**Claims And Evidence:**

Yes

**Claims Explanation:**

All reviewers agree that the paper clearly meets the criterion related to "Claims And Evidence" by providing claims that are sound and well supported by evidence. The reviewers raised some questions and concerns, which the authors have adequately addressed in the revisions to the reviewers' satisfaction.